# Biophysical network modeling of temporal and stereotyped sequence propagation of neural activity in the premotor nucleus HVC

**Zeina Bou Diab, Marc Chammas, Arij Daou***

Neurophysiology and Computational Neuroscience Group, Biomedical Engineering Program, American University of Beirut, Beirut, Lebanon

## eLife Assessment

This computational study examines how neurons in the songbird premotor nucleus HVC might generate the precise, sparse burst sequences that drive adult song. The findings would be **useful** for understanding how intrinsic conductances and HVC microcircuitry may produce neural sequences, but the work is **incomplete** because of arbitrary network assumptions, insufficient consideration of biological details such as how silent gaps in song sequences are represented, and failure to incorporate interactions with auditory and brainstem inputs. As a result, the study offers limited advance and only a modest conceptual advance over prior models.

***For correspondence:**
arij.daou@aub.edu.lb

**Competing interest:** The authors declare that no competing interests exist.

**Abstract** Stereotyped neural sequences are often exhibited in the brain, yet the neurophysiological mechanisms underlying their generation are not fully understood. Birdsong is a prominent model to study such behavior, particularly because juvenile songbirds progressively learn from their tutors and by adulthood are able to sing stereotyped song patterns. The songbird premotor nucleus HVC coordinates motor and auditory activity responsible for learned vocalizations. The HVC comprises three neural populations that have distinct in vitro and in vivo electrophysiological responses. Typically, models that explain HVC's network either rely on intrinsic HVC circuitry to propagate sequential activity, rely on extrinsic feedback to advance the sequence, or rely on both. Here, we developed a physiologically realistic neural network model incorporating the three classes of HVC neurons based on the ion channels and the synaptic currents that had been pharmacologically identified. Our model is based on a feedforward chain of microcircuits that encode for the different subsyllabic segments (SSSs) and that interact with each other through structured feedback inhibition. The network reproduced the in vivo activity patterns of each class of HVC neurons and unveiled key intrinsic and synaptic mechanisms that govern the sequential propagation of neural activity by highlighting important roles for the T-type $Ca^{2+}$ current, $Ca^{2+}$-dependent $K^+$ current, A-type $K^+$ current, hyperpolarization-activated inward current, as well as excitatory and inhibitory synaptic currents. The result is a biophysically realistic model that suggests an improved characterization of the HVC network responsible for song production in the songbird.

## Introduction

Learned temporal sequences are expressed in many brain regions and play a critical role in temporal information encoding, such as navigation (*Skaggs et al., 1996*), the timing of motor actions (*Wang et al., 2018*), time generation (*MacDonald et al., 2011*; *Pastalkova et al., 2008*), decision making

**Figure 1.** Song system overview and general intrinsic and network properties of HVC neurons in vivo and in vitro. (**A**) Schematic diagram showing a sagittal view of the male zebra finch song system. The vocal motor pathway (VMP, red color) contains circuits that directly pattern song output. The anterior forebrain loop (AFP, blue color) pathway contains circuits that are important for song learning and plasticity. (**B**) HVC includes multiple classes of neurons; HVC$_X$ neurons that project to area X (blue), HVC$_{RA}$ neurons that project to nucleus RA (red), and HVC interneurons (HVC$_{INT}$, black). HVC$_X$ and HVC$_{RA}$ excite HVC$_{INT}$ via AMPA and NMDA synapses (green arrow), while HVC$_{INT}$ neurons inhibit both classes of projecting neurons via GABA synapses (brown arrows with circle heads). Each class of HVC neurons is characterized by its own family of ionic currents (*Daou et al., 2013*). (**C**) HVC$_{RA}$ neurons exhibit a very sparse activity during singing eliciting a single 4–6 ms burst at a single and exact moment in time during each rendition of the song. On the contrary, HVC interneurons burst densely throughout the song (Adapted from *Hahnloser et al., 2002*). (**D**) Similar to HVC$_{RA}$, HVC$_X$ neurons generate 1–4 bursts that are time-locked and highly stereotyped from one rendition of the song to another (Adapted from *Kozhevnikov and Fee, 2007*).

© 2002, Nature. Figure 1C is reprinted from Figure 2B from *Hahnloser et al., 2002*, with permission from Nature. It is not covered by the CC-BY 4.0 licence and further reproduction of this panel would need permission from the copyright holder.

© 2007, American Physiological Society. Figure 1D is reprinted from Figure 2A from *Kozhevnikov and Fee, 2007*, with permission from American Physiological Society. It is not covered by the CC-BY 4.0 licence and further reproduction of this panel would need permission from the copyright holder.

(*Harvey et al., 2012*; *Schmitt et al., 2017*), and skilled movement (*Peters et al., 2014*). Zebra finches sing remarkably stereotyped songs, rendering them as an excellent model for studying the mechanisms underlying neural sequences (*Hahnloser et al., 2002*). The anatomical basis for song production and learning is a highly developed neural network known as the song system (*Figure 1A*), with nucleus HVC exhibiting a rhythmic pattern-generating role encoding for the syllable order and the overall temporal structure of the birdsong (*Fee and Goldberg, 2011*; *Fee et al., 2004*; *Fee and Scharff, 2010*; *Mooney, 2009*; *Yu and Margoliash, 1996*).

There are three main neuronal populations in the HVC exhibiting different functional, cellular, and pharmacological properties (*Daou et al., 2013*; *Kubota and Taniguchi, 1998*; *Mooney, 2000*; *Mooney and Prather, 2005*): neurons that project to the robust nucleus of arcopallium (RA; $HVC_{RA}$), neurons that project to Area X ($HVC_X$), and interneurons ($HVC_{INT}$, *Figure 1B*). $HVC_{Av}$ neurons (projecting to nucleus Avalanche) exist and play a role in song learning, yet their intrinsic and synaptic properties remain unknown (*Roberts et al., 2017*). Intracellular recordings showed that $HVC_{RA}$, $HVC_X$, and $HVC_{INT}$ neurons have distinctive in vivo and in vitro electrophysiological properties (*Daou et al., 2013*; *Dutar et al., 1998*; *Kubota and Taniguchi, 1998*; *Mooney et al., 2001*; *Shea et al., 2010*), which are orchestrated via a family of ion channels (*Daou et al., 2013*).

During singing, $HVC_{RA}$ neurons produce a single burst of spikes (~10 ms) that is tightly locked to the song (*Figure 1C*, *Hahnloser et al., 2002*; *Kozhevnikov and Fee, 2007*). $HVC_X$ neurons, in their turn, elicit 1–4 bursts that are also time-locked to vocalizations (*Figure 1D*, *Kozhevnikov and Fee, 2007*), while $HVC_{INT}$ neurons exhibit tonic activation (*Figure 1C*), with bursting and suppression at different locations throughout the song (*Amador et al., 2013*; *Cannon et al., 2015*; *Hahnloser et al., 2002*; *Kosche et al., 2015*; *Kozhevnikov and Fee, 2007*; *Long et al., 2010*; *Markowitz et al., 2015*).

Several models of how sequence is generated within HVC have been proposed (*Cannon et al., 2015*; *Drew and Abbott, 2003*; *Egger et al., 2020*; *Elmaleh et al., 2021*; *Galvis et al., 2018*; *Gibb et al., 2009a*; *Hamaguchi et al., 2016*; *Jin, 2009*; *Long and Fee, 2008*; *Markowitz et al., 2015*). These models either rely on intrinsic HVC circuitry to propagate sequential activity, rely on extrinsic feedback to advance the sequence, or rely on both. The proposed models do not capture the complex details of spike morphology, do not include the right ionic currents, do not incorporate all classes of HVC neurons, or do not generate realistic firing patterns as seen in vivo. In this work, we ask a simple but powerful question: Can the biologically realistic intrinsic properties of HVC neurons and the local synaptic connections among them produce the precise sequence propagation seen in birdsong? We are particularly interested in understanding what core biophysical ingredients (ionic and synaptic currents) are truly necessary to generate sparse sequences while preserving each HVC neuron activity in vivo. How do features like bursting behavior of these neurons or patterns of inhibition help shape the flow of activity throughout the network? To address these questions, we developed a physiologically realistic network model incorporating the three classes of HVC neurons based on the ion channels and the synaptic currents that had been pharmacologically identified (*Daou et al., 2013*; *Kosche et al., 2015*; *Mooney and Prather, 2005*). Our model is based on a feedforward chain of microcircuits that encode for the different sub-syllabic segments (SSSs) and that interact with each other through structured feedback inhibition. The network developed unveiled key intrinsic and synaptic mechanisms that govern the sequential propagation of neural activity by highlighting important roles for the T-type $Ca^{2+}$ current, $Ca^{2+}$-dependent $K^+$ current, A-type $K^+$ current, hyperpolarization-activated inward current, as well as excitatory and inhibitory synaptic currents. Our model provides a new way of thinking about sequence generation during birdsong vocalizations and in network architectures more generally.

## Results

Adult zebra finches generate intricate songs composed of sequences of distinct song elements, each characterized by a stereotypical acoustic pattern across every rendition of song. The neural circuitry that governs this behavior consists of $HVC_{RA}$ neuronal population, each of which emits a single and stereotyped 6–10ms burst during each rendition of song, $HVC_X$ neurons eliciting 1–4 bursts that are similarly time-locked to vocalizations, and $HVC_{INT}$ neurons that tend to burst densely throughout song. Intrinsic and synaptic mechanisms that orchestrate these neurons' behaviors are well known (*Daou et al., 2013*; *Kornfeld et al., 2017*; *Mooney and Prather, 2005*). We next describe the steps of building our biophysical network model to describe this ongoing behavior in the following order: (1)

tuning the synaptic parameters to fit the dual-intracellular recording traces collected experimentally by *Mooney and Prather, 2005* as well as *Kosche et al., 2015*, and then (2) describing the network components that are essential for the patterned output of the system, as well as the internal dynamics of the network that governs the strength and duration of individual bursts, the duration of silent gaps between bursts, sparseness versus tonicity, the interplay between excitation and inhibition, role of intrinsic properties, and so on that explain how the firing activity of the three classes of HVC neurons propagates through the network in a sequential manner.

## Tuning synaptic parameters

We initiated our HVC network modeling study calibrating the synaptic parameters (excitatory and inhibitory currents' activation/inactivation constants, etc.) by reproducing the voltage traces elicited by the dual intracellular recordings from identified pairs of HVC neurons in brain slices conducted by *Mooney and Prather, 2005*. While we are using off-the-shelf synaptic currents from the literature (*Destexhe et al., 1994*; *Varela et al., 1997*), we needed to make sure that the synaptic parameters used could replicate the dual synaptic connectivity patterns (strengths of excitation/inhibition, magnitudes of voltage deflections, and other trace morphologies). Mooney and Prather's findings revealed robust disynaptic feedforward inhibition from $HVC_{RA}$ to $HVC_X$ neurons (mediated by $HVC_{INT}$ neurons), potent monosynaptic excitation from $HVC_{RA}$ and $HVC_X$ to $HVC_{INT}$ neurons (via NMDA and AMPA currents), and substantial monosynaptic inhibition from $HVC_{INT}$ neurons to $HVC_{RA}$ and $HVC_X$ (via GABA currents).

*Figure 2* displays the dual intracellular recordings conducted by Mooney and Prather (left column) as well as the mathematical model replications (right column) after the synaptic parameters' calibration. DC-evoked action potentials in $HVC_{RA}$ neurons trigger inhibitory postsynaptic potentials (iPSPs) in $HVC_X$ neurons (*Figure 2-A1*), as well as fast depolarizing postsynaptic potentials (dPSPs) in $HVC_{INT}$ neurons (*Figure 2-B1*). To replicate the effects of stimulating $HVC_{RA}$ neurons onto the other two classes of HVC neurons, we connected one $HVC_{RA}$ neuron to excite one $HVC_{INT}$ neuron via an AMPA current. The $HVC_{INT}$ neuron, in turn, was connected with one $HVC_X$ neuron via a GABA current, thereby making a di-synaptic pathway from $HVC_{RA}$ to $HVC_X$. DC-evoked action potentials in the model $HVC_{RA}$ neuron (brief ~10ms depolarizing current pulses 0.5 nA, similar to what Mooney and Prather applied to $HVC_{RA}$ neurons) evoked a fast-depolarizing postsynaptic potential (dPSP) in the corresponding model $HVC_{INT}$ neuron (*Figure 2-B2*) as well as inhibitory postsynaptic potential (iPSP) in the corresponding model $HVC_X$ neuron (*Figure 2-A2*) mediated via $HVC_{INT}$. Similarly, DC-evoked action potentials in $HVC_{INT}$ neurons generate fast iPSPs in $HVC_X$ neurons (*Figure 2-C1-D1*). The unequal magnitude of the four sags elicited in the $HVC_X$ neuron (due to the four brief stimuli) and the large sag after the stimuli ends (*Figure 2-C1*), as well as the jagged long sag in response to the repetitive action potentials elicited by the $HVC_{INT}$ neuron (*Figure 2-D1*), are probably due to the fact that the corresponding $HVC_X$ neurons are receiving multiple synaptic inputs from neurons other than the neurons being stimulated by Mooney and Prather, which adds to the underlying nonlinearities of the responses being recorded. Model $HVC_{INT}$ and $HVC_X$ neurons were connected and synaptic parameters were calibrated to generate similar waveforms by giving brief ~10ms depolarizing current pulses of 0.5 nA to $HVC_{INT}$ (*Figure 2-C2*), or giving a DC-pulse of 0.5 nA for 500 ms (*Figure 2-D2*). In the model, the sag seen in the $HVC_X$ neuron response is due to the build-up of the H-current as the model $HVC_{INT}$ neurons continue firing, exerting its inhibition onto the $HVC_X$ neuron. Finally, model $HVC_X$-$HVC_{INT}$ monosynaptic connectivity (*Figure 2-E2*) was calibrated to match the experimental findings (*Figure 2-E1*).

## Network architecture: non-sequential random sampling in HVC

We next describe the activity patterns generated by our network model and explain how the firing activity of the three classes of HVC neurons propagates through the network in sequential bursts of activity. The network developed is composed of chains of HVC subnetworks or microcircuits, each with its own intrinsic dynamics. The microcircuit represents a basic architectural unit that encodes for a syllable or an SSS in the motif (*Figure 3*). Each neuron in a microcircuit is representative of a neural population. In other words, a model neuron (belonging to any class) firing is representative of a population of that neuronal class firing, which could be many neurons of the same class exhibiting very similar intrinsic and synaptic properties leading to their firing at the same time. We refer here to 'microcircuits' in a more functional sense, rather than rigid, isolated spatial divisions (*Cannon et al.,*

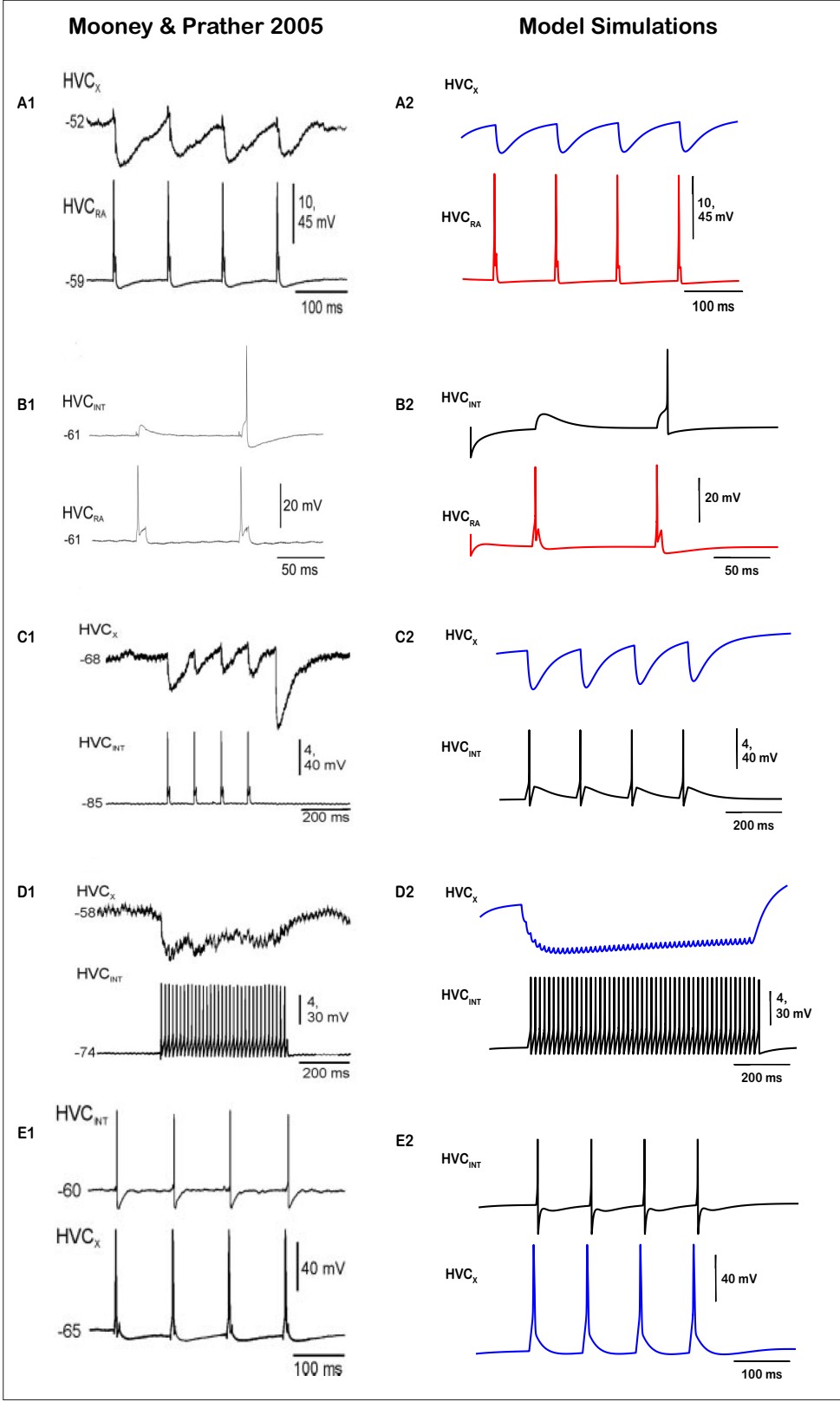

**Figure 2.** Model output compared to experimental results obtained by *Mooney and Prather, 2005*. DC-evoked action potentials in HVC$_{RA}$ neurons trigger iPSPs in HVC$_X$ neurons (**A1**) as well as fast dPSPs in HVC$_{INT}$ neurons (**B1**). Brief (~10 ms) depolarizing current pulses (0.5 nA) applied to model HVC$_{RA}$ neuron (same values used as by *Mooney and Prather, 2005*) evoke similar responses in the corresponding model HVC$_X$ (**A2**) and model HVC$_{INT}$

*Figure 2 continued*

(**B2**) neurons. DC-evoked action potentials in HVC$_{INT}$ neurons generate fast iPSPs in HVC$_X$ neurons (**C1, D1**). Similar responses were elicited in model HVC$_X$ neurons when model HVC$_{INT}$ neuron was stimulated by brief (10 ms) depolarizing pulses (0.5 nA) (**C2**) or when it was given a DC-pulse of 0.5 nA for 500ms (**D2**). Finally, HVC$_{INT}$ neurons elicit fast dPSPs when HVC$_X$ neurons are injected with 10 ms pulses of 0.5 nA current (**E1**), which was simulated in the model (**E2**). In this and subsequent figures (unless otherwise specified), HVC$_X$ neurons' traces are represented in blue, HVC$_{RA}$ neurons in red, and HVC$_{INT}$ neurons in black. Panels in the left column are adapted from ***Mooney and Prather, 2005***.

The online version of this article includes the following source code for figure 2:

**Source code 1.** MATLAB code that contains the underlying ODEs and parameters for the individual neurons connected in the network.

**Source code 2.** MATLAB code that contains all underlying ODEs, equations and corresponding parameters for the interneurons used to simulate this network.

**Source code 3.** MATLAB code that contains all underlying ODEs, equations and corresponding parameters for the X-projecting neuron used to simulate this network.

**Source code 4.** MATLAB code that contains all underlying ODEs, equations and corresponding parameters for the RA-projecting neuron used to simulate this network.

---

*2015*). A microcircuit in our model reflects the local rules that govern the interaction between all HVC neuron classes within the broader network and that are essential for proper activity propagation. The number of microcircuits in the chain determines the number of SSSs in the motif. We envision the HVC to be composed of many copies of such microcircuit chains that are associated with SSSs with

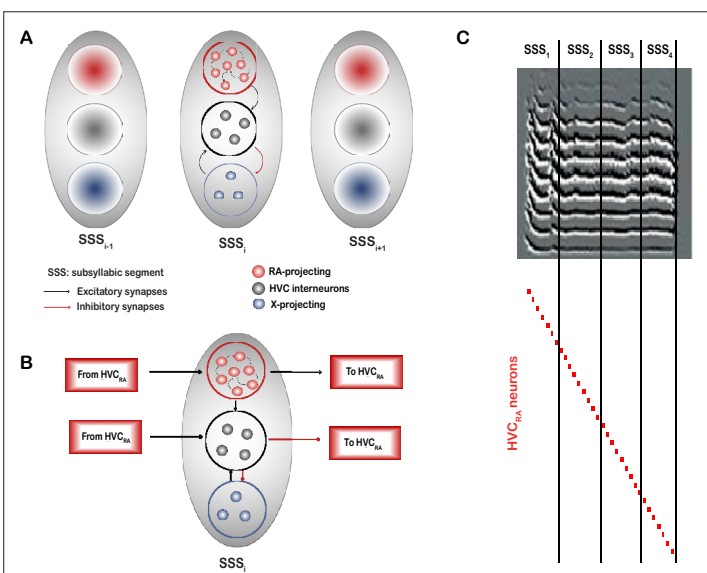

**Figure 3.** Cartoon diagram illustrating the network architecture configuration. (**A**) Each gray oval represents a microcircuit encoding for a sub-syllabic segment (SSS). The number of microcircuits is envisioned to be equal to the number of SSSs representing the song. Each microcircuit contains a number of HVC$_{RA}$, HVC$_X$, and HVC$_{INT}$ neurons selected randomly from the total pool of neurons (see text). (**B**) Within each microcircuit, HVC$_{RA}$ neurons are connected to each other in a chain-like mode and they send excitatory afferents to HVC$_{INT}$ neurons in the same and other microcircuits, selected randomly. HVC$_{INT}$ neurons send GABAergic synapses onto HVC$_X$ neurons in the same microcircuit only as well as to HVC$_{RA}$ neurons in any other random microcircuit except the microcircuit they belong to. Finally, HVC$_X$ neurons send excitatory afferents to HVC$_{INT}$ neurons in the same microcircuit. Activity starts by a small DC pulse to the first HVC$_{RA}$ neuron in the first microcircuit. Activity propagates from one microcircuit to another by excitatory coupling between the last HVC$_{RA}$ neuron in microcircuit i and the first HVC$_{RA}$ neuron in microcircuit i+1. (**C**) During singing, the propagation of activity unfolds across the chain of microcircuits, such that neurons belonging to microcircuit x get activated and encode for SSS$_x$.

roughly synchronized activity. The duration of the SSSs need not be the same; therefore, the number of neurons that each microcircuit encompasses need not be equal as we will see next. Moreover, while silent gaps are integral to the overall process of song production, we have not elaborated on them in this model due to the lack of a clear, biophysically grounded representation for the gaps themselves at the level of HVC. Our primary focus here is on modeling the active, syllable-producing phases of the song, where the HVC network's sequential dynamics are critical for song. However, one can think of the encoding of silent gaps via similar mechanisms that encode SSSs, where each gap is encoded by similar microcircuits comprised of the three classes of HVC neurons (let's call them GAP, rather than SSS) that are active only during the silent gaps. In this case, the propagation of sequential activity is carried throughout the GAPs from the last SSS of the previous syllable to the first SSS of the subsequent syllable.

The network is comprised of a total pool of $HVC_{RA}$ neurons (120 neurons, red circles), a pool of interneurons (50 neurons, black circles) and a pool of $HVC_X$ neurons (50 neurons, blue circles), thereby maintaining a 2:1:1 proportionality factor across the populations of $HVC_{RA}$: $HVC_{INT}$: $HVC_X$ as reported earlier (**Kornfeld et al., 2017**; **Wild et al., 2005**). The total number of neurons in the pool is arbitrary and can be made larger, but we limited it to these values given the huge number of ODEs that are being simulated (~2000 ODEs). **Figure 3A** shows the network diagram illustrating three sample microcircuits encoding for three SSSs ($SSS_{i-1}$, $SSS_i$, and $SSS_{i+1}$) in sequence. Each microcircuit (enclosed by a gray oval) is made up of a number of $HVC_{RA}$ neurons (red circles) and a number of $HVC_{INT}$ neurons (black circles), all selected randomly from their corresponding pools, as we will describe next. In our model, we limited the number of microcircuits to twenty (i.e. the motif is encoded by 20 SSSs), but this number is also arbitrary and can be made larger or smaller.

## Network organization

The total pool of $HVC_{RA}$ neurons is comprised of smaller groups of $HVC_{RA}$ neurons, the number in each group of which is chosen randomly from the pool (red background circles in each gray oval). Each group of $HVC_{RA}$ neurons belongs to a unique microcircuit, and no $HVC_{RA}$ neuron is allowed to be part of more than one microcircuit for reasons described next. Since we set the motif to be represented by 20 microcircuits in our illustration, $HVC_{RA}$ neurons were recruited to their corresponding microcircuits randomly, with each microcircuit allowing a random number (3 - 10) neurons of the RA-projectors to belong to it. In addition to that, the numbers of $HVC_{INT}$ neurons (black background circles) that a microcircuit exhibits are random numbers between 1 and 4, as well as the number of $HVC_X$ neurons (blue background circles) in each microcircuit is random between 1 and 4, where the individual neurons are selected arbitrarily one neuron at a time from their corresponding pools. Each $HVC_{INT}$ neuron can belong to a single microcircuit, and similarly, each $HVC_X$ neuron can belong to a single microcircuit for reasons described next.

## Synaptic connectivity

Within each microcircuit, $HVC_{RA}$ neurons are selected randomly, one after the other, to send AMPA excitatory synapses to each other in a chain-like mode (**Figure 3B**). Specifically, if there are $m$ $HVC_{RA}$ neurons recruited to belong to microcircuit $i$ (where $m$ is the random number generated between 3 and 10 in this case), a neuron from the $m$ is first selected randomly and designated as the first neuron in the chain ($HVC_{RA_1}^i$). After that, a second neuron ($HVC_{RA_2}^i$) from the remaining $m-1$ is selected randomly and an AMPA synapse is connected from $HVC_{RA_1}^i$ to $HVC_{RA_2}^i$. Similarly, a third neuron ($HVC_{RA_3}^i$) is selected randomly from the remaining $m-2$ neurons and an AMPA synapse is connected from $HVC_{RA_2}^i$ to $HVC_{RA_3}^i$, and so on, until the $HVC_{RA}$ neurons in every microcircuit are connected together (**Figure 3A–B**, small red circles in each microcircuit).

Each $HVC_{INT}$ in a microcircuit was assigned a random number (between 3 and 8) of excitatory AMPA connections from the $HVC_{RA}$ neurons in the same microcircuit it belongs to, as well as from $HVC_{RA}$ neurons in the other microcircuits (**Figure 3B**). In their turn, each $HVC_{INT}$ neuron sends a random number (between 2 and 4) of GABAergic inputs to $HVC_{RA}$ neurons, chosen arbitrarily from any microcircuit in the chain except the microcircuit that the $HVC_{INT}$ neuron belongs to, due to the following reason: if $HVC_{INT}$ inhibits $HVC_{RA}$ in the same microcircuit, some of the $HVC_{RA}$ bursts in the microcircuit might be silenced by the dense and strong $HVC_{INT}$ inhibition breaking the chain of activity. However, if $HVC_{INT}$ inhibits $HVC_{RA}$ in any other microcircuit, activity is ensured to propagate because the $HVC_{INT}$

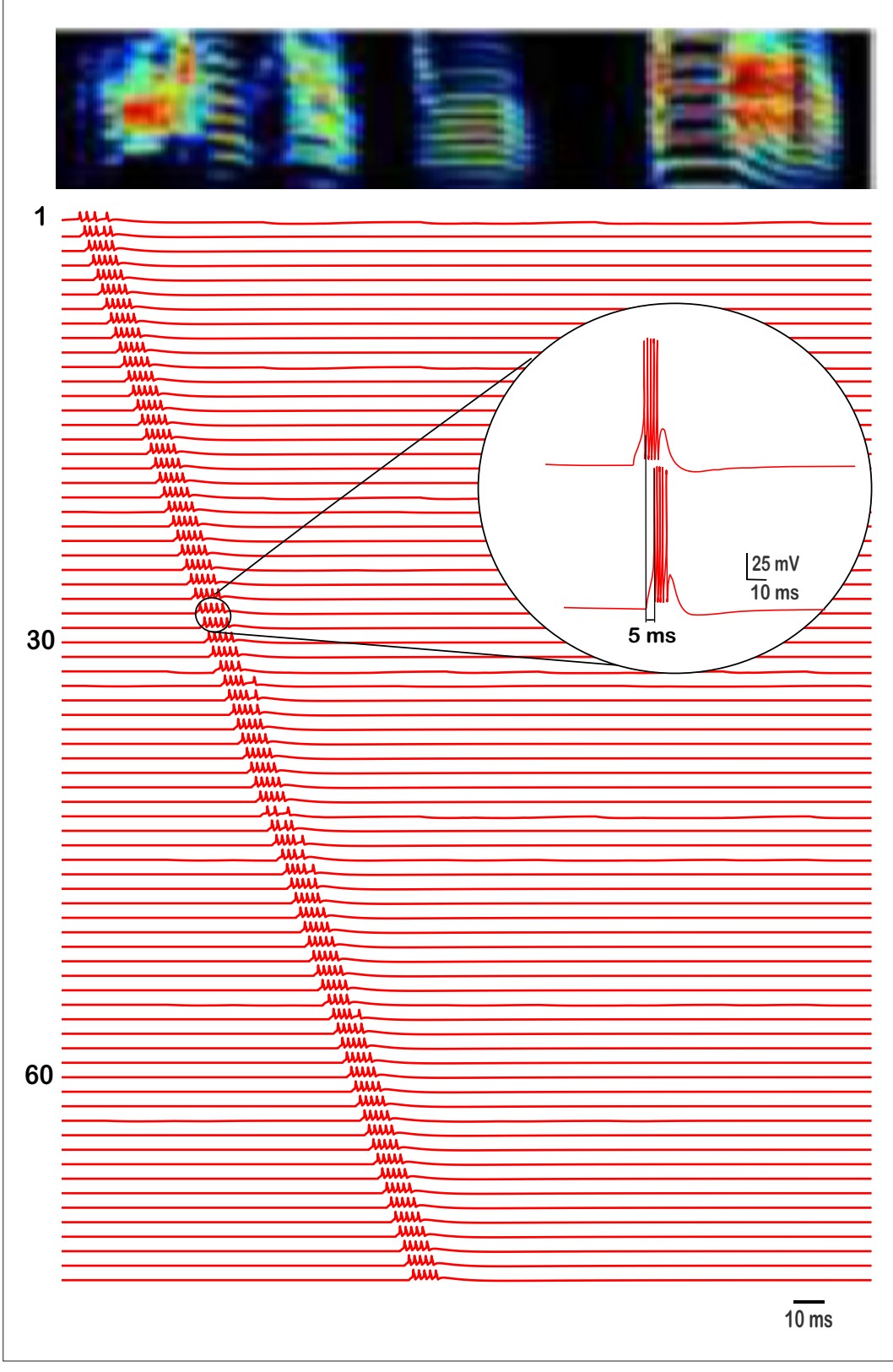

**Figure 4.** Spiking patterns of 120 HVC$_{RA}$ neurons (labeled with numbers) showing the propagation of sequential activity. The neural traces are aligned by the acoustic elements of a spectrogram from an exemplar bird's song illustrating the firing of HVC$_{RA}$ neurons with respect to ongoing part of a song. The inset shows a zoomed version of two subsequent HVC$_{RA}$ neurons firing patterns illustrating the delay between their individual bursts.

inhibition of the corresponding HVC$_{RA}$ would arrive at times that are not the 'assigned' times of the HVC$_{RA}$ to elicit their ultra-sparse code (*Hahnloser et al., 2002*).

Unlike HVC$_{RA}$, HVC$_{INT}$ neurons belonging to a particular microcircuit can burst at times other than the moments when the corresponding encoded SSS is being 'sung'; however, we chose to house interneurons within microcircuits for the mere fact that any given interneuron cannot inhibit any given HVC$_{RA}$ neuron; rather, there are some 'rules of engagement' where we ensure that no inhibition arrives to any HVC$_{RA}$ neuron while it's eliciting its burst of activity (*Figure 3B*). In other words, what makes a particular interneuron belong to this microcircuit or the other is merely the fact that it cannot inhibit HVC$_{RA}$ neurons that are housed in the microcircuit it belongs to for the reasons described. In this regard, this arrangement is similar to the *Cannon et al., 2015* model in the context of structured inhibition amid the ongoing feedforward excitation.

At the HVC$_X$ side, each X-projecting neuron excites via AMPA currents a random selection (1 - 3) of HVC$_{INT}$ neurons that belong to the same microcircuit, and in their turn, each HVC$_{INT}$ neuron inhibits via GABA synapses a random selection (1 - 2) of HVC$_X$ neurons in the same microcircuit (*Figure 3B*). These numbers are again arbitrary, but we limited the number of connections from HVC$_{INT}$ to HVC$_X$ due to the fact that X-projecting neurons in our model fire upon rebound from inhibition, and the more inhibitory inputs they receive, the more rebound bursts they elicit, which is not realistic since HVC$_X$ neurons are known to elicit 1–4 bursts during singing (*Fujimoto et al., 2011*; *Kozhevnikov and Fee, 2007*), which is what we achieved with this number of synapses. HVC$_X$ neurons were selected to be housed within microcircuits and their synapses connecting to interneurons within the same microcircuit due to the following reason: if an HVC$_X$ neuron belonging to microcircuit $i$ sends excitatory input to an HVC$_{INT}$ neuron in microcircuit $j$, and that interneuron happens to select an HVC$_{RA}$ neuron from microcircuit $i$ as its afferent inhibitory connection (via random sampling), then the propagation of sequential activity will halt, and we'll be in a scenario similar to what was described earlier for HVC$_{INT}$ neurons inhibiting HVC$_{RA}$ neurons in the same microcircuit. Similarly, if an HVC$_{INT}$ neuron in microcircuit $i$ inhibits an HVC$_X$ neuron in another microcircuit $j$, and that HVC$_X$ neuron excites an interneuron that synapses onto an HVC$_{RA}$ from microcircuit $i$, then sequential activity might be disrupted.

While HVC$_{RA}$ neurons are connected to each other in each microcircuit in a chain-like mode as described earlier, the microcircuits interact with each other via the projections from the last HVC$_{RA}$ in a microcircuit $i$ to the first HVC$_{RA}$ in a following microcircuit $i + 1$. The network is kick-started by a stochastic DC input to $HVC_{RA}^1$, that is, only $HVC_{RA}^1$ receives input from outside HVC. During singing, the propagation of activity unfolds across the chain of microcircuits, such that neurons belonging to microcircuit $x$ get activated and encode for *SSSx* (*Figure 3C*). The propagation of sequential activity along with the realistic firing of the three classes of HVC neurons is maintained and orchestrated by HVC's intrinsic and synaptic processes without relying on extrinsic inputs as shown next (*Figure 4*, *Figure 5*, *Figure 6*, *Figure 7*, *Figure 8*, *Figure 9*, *Figure 10*).

## Sequential propagation of HVC$_{RA}$ activity

The activity patterns that RA-projecting neurons of this network display are illustrated in *Figure 4*. HVC$_{RA}$ neurons burst extremely sparsely, generating at most a single burst per simulation (song motif) and with different HVC$_{RA}$ neurons bursting at different times in the song. HVC$_{RA}$ bursts had a duration of 8.12±0.89ms, and comprised of 4.76±0.48 spikes. The inset of *Figure 4* shows the delay from the onset of the first spike in an HVC$_{RA}$ neuron to the onset of the first spike in the next HVC$_{RA}$ it synapses to. This duration is controlled by two factors: (1) the magnitude of the AMPA conductance, with larger magnitudes corresponding to shorter delays and vice versa (*Figure 5A*). In short, the faster the AMPA current is (modeled as larger magnitude in the $g_{AMPA}$ parameter that connects two HVC$_{RA}$ neurons), the shorter the delay between the successive HVC$_{RA}$ bursts. (2) The magnitude of the A-type K$^+$ current conductance ($g_A$) with larger magnitudes corresponding to longer delays (*Figure 5B*). This conductance increases rapidly on depolarization due to $I_A$'s fast activation. The rapid increase in $g_A$ halts after a few milliseconds and switches to a slow decrease that is due to the slow inactivation that this current exhibits. The slow decrease is reflected in the voltage trace as a slow depolarization in the membrane potential (encoding for the delay to spiking), and this allows the model HVC$_{RA}$ neuron to eventually escape the inhibition produced by $I_A$ and fire its delayed burst.

The number of spikes in HVC$_{RA}$ neurons and the strength of the burst is controlled by three factors: (1) the strength of the AMPA conductance itself where stronger excitatory coupling corresponds

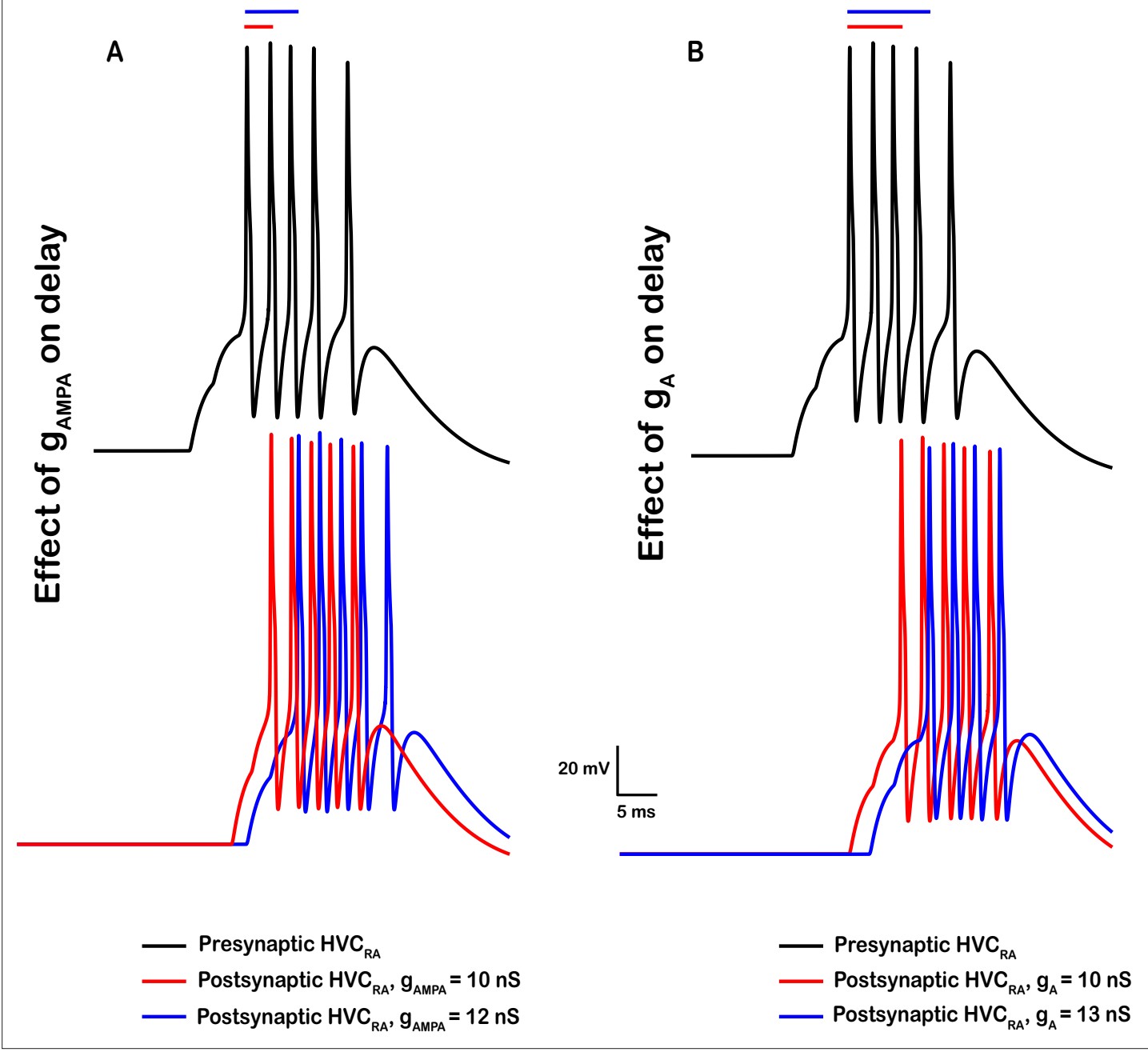

**Figure 5.** Effects of the AMPA synaptic conductance and A-type K$^+$ conductance on the delay between two successive HVC$_{RA}$ bursts. (**A**) Presynaptic model HVC$_{RA}$ neuron (top, black) is connected to a postsynaptic model HVC$_{RA}$ neuron and the corresponding AMPA excitatory conductance ($g_{AMPA}$) was increased from 10 nS (bottom, red) to 12 nS (bottom, blue), while keeping all other parameters fixed. Increasing $g_{AMPA}$ reduces the delay between the peaks of the pre- and post- HVC$_{RA}$'s first spikes and increases the number of spikes in the postsynaptic neuron. (**B**) Larger magnitudes of the A-type K$^+$ conductance ($g_A$) leads to longer delays to spiking. While keeping all intrinsic and synaptic parameters fixed ($g_{AMPA}$ +10), increasing $g_A$ from 10 nS (bottom, red) to 13 nS (bottom, blue) delayed the onset to spiking and reduced the number of spikes. Bars on the top show the duration in ms between the peak of the first action potential in the presynaptic neuron to the peak of the first action potential in the postsynaptic neuron.

to larger number of spikes (*Figure 5A*), (2) the A-type K$^+$ conductance where larger magnitudes of the conductance reduce the general excitability of the HVC$_{RA}$ neuron and limit its number of spikes (*Figure 5B*), and (3) the interplay between the L-type Ca$^{2+}$ ($g_{CaL}$) and the Ca$^{2+}$-dependent K$^+$ ($g_{SK}$) conductances that control the strong adaptation which these neurons exhibit (*Daou et al., 2013*). In short, intracellular Ca$^{2+}$ (due to $I_{CaL}$) accumulates during the HVC$_{RA}$ burst. This results in a buildup of

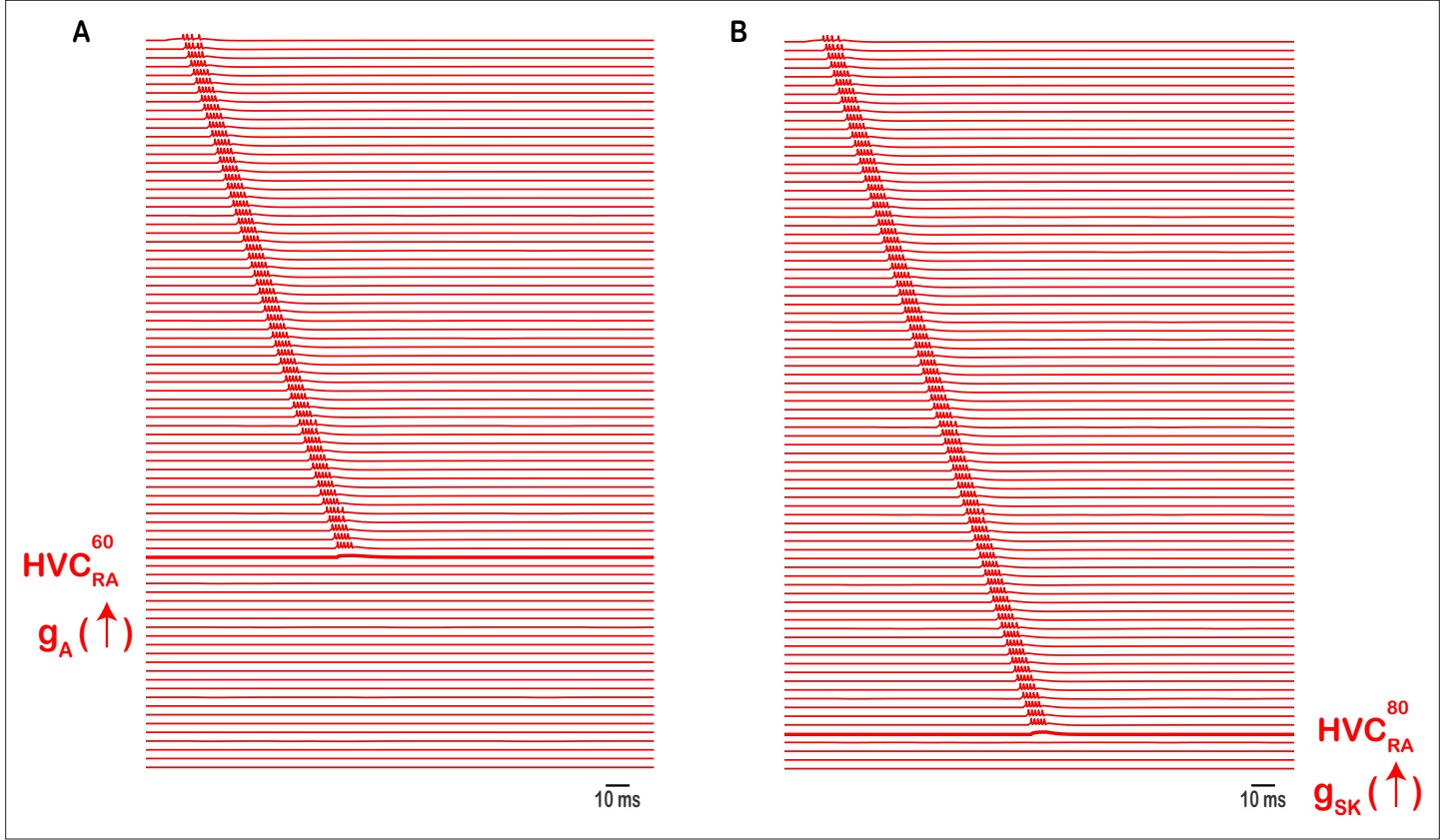

**Figure 6.** Intrinsic changes in HVC$_{RA}$ halt the propagation of sequential activity. Up-regulating the A-type K$^+$ current (**A**) or the Ca$^{2+}$ - dependent K$^+$ current (**B**) in exemplar neurons ($HVC_{RA}^{60}$, **A**) or ($HVC_{RA}^{80}$, **B**), by increasing $g_A$ + (**A**), or $g_{SK}$ from 15-fold nS (**B**), reduces the excitability of corresponding HVC$_{RA}$ neuron markedly, eliminating its corresponding burst and breaking the sequence.

Ca$^{2+}$-activated K$^+$ current ($I_{SK}$) that terminates the HVC$_{RA}$ burst, which in turn terminates any burst in a post-synaptic neuron since HVC$_{RA}$ can no longer provide excitation.

The propagation of sequential bursting in HVC$_{RA}$ neurons halts, and the chain of HVC$_{RA}$ activity is broken if any of the following is satisfied: (1) if an AMPA conductance for any of the HVC$_{RA}$'s that connects it to the next HVC$_{RA}$ in its chain is small enough such that it's not able to elicit sufficient excitability on the postsynaptic side, (2) if a $g_A$ conductance (**Figure 6A**) or a $g_{SK}$ conductance (**Figure 6**) in any of the HVC$_{RA}$'s is large enough (mimicking an up-regulation in the corresponding channel) to eliminate the corresponding HVC$_{RA}$'s burst. Therefore, in order to generate accurate HVC$_{RA}$ bursting patterns that maintain the sequential propagation of neural activity, accurate number of spikes and delays between spikes across the population of HVC$_{RA}$'s in all microcircuits, as well as the general intrinsic properties of the HVC$_{RA}$ neurons' spike morphologies (see Materials and methods), both intrinsic and synaptic constraints are needed with key roles in this process for the A-type K$^+$ and the Ca$^{2+}$-dependent K$^+$ currents that HVC$_{RA}$ model neurons exhibit, as well as the AMPA currents connecting the HVC$_{RA}$ population together within and across microcircuits. Recall that we envision each HVC neuron of any class in our model as a representative of a neural population of the same class that exhibits the same intrinsic as well as afferent and efferent synaptic connectivity. Therefore, in **Figure 6**and the subsequent figures (9, 11, and 12) where we show disruption of sequential activity due to changes in synaptic or intrinsic properties of HVC neurons, the modeled synaptic/intrinsic changes to the selected neurons shown are envisioned to be changes applied to the whole neural population encoded by our model neuron. In other words, disrupting the properties of a single neuron within that neural population will not cause harm to the propagation of activity due to what could be thought of as homeostatic mechanisms of the network (**Golowasch et al., 1999**; **Marder and Goaillard, 2006**; **Williams et al., 2013**). This redundancy within the population allows the propagation of activity to be maintained. It is an

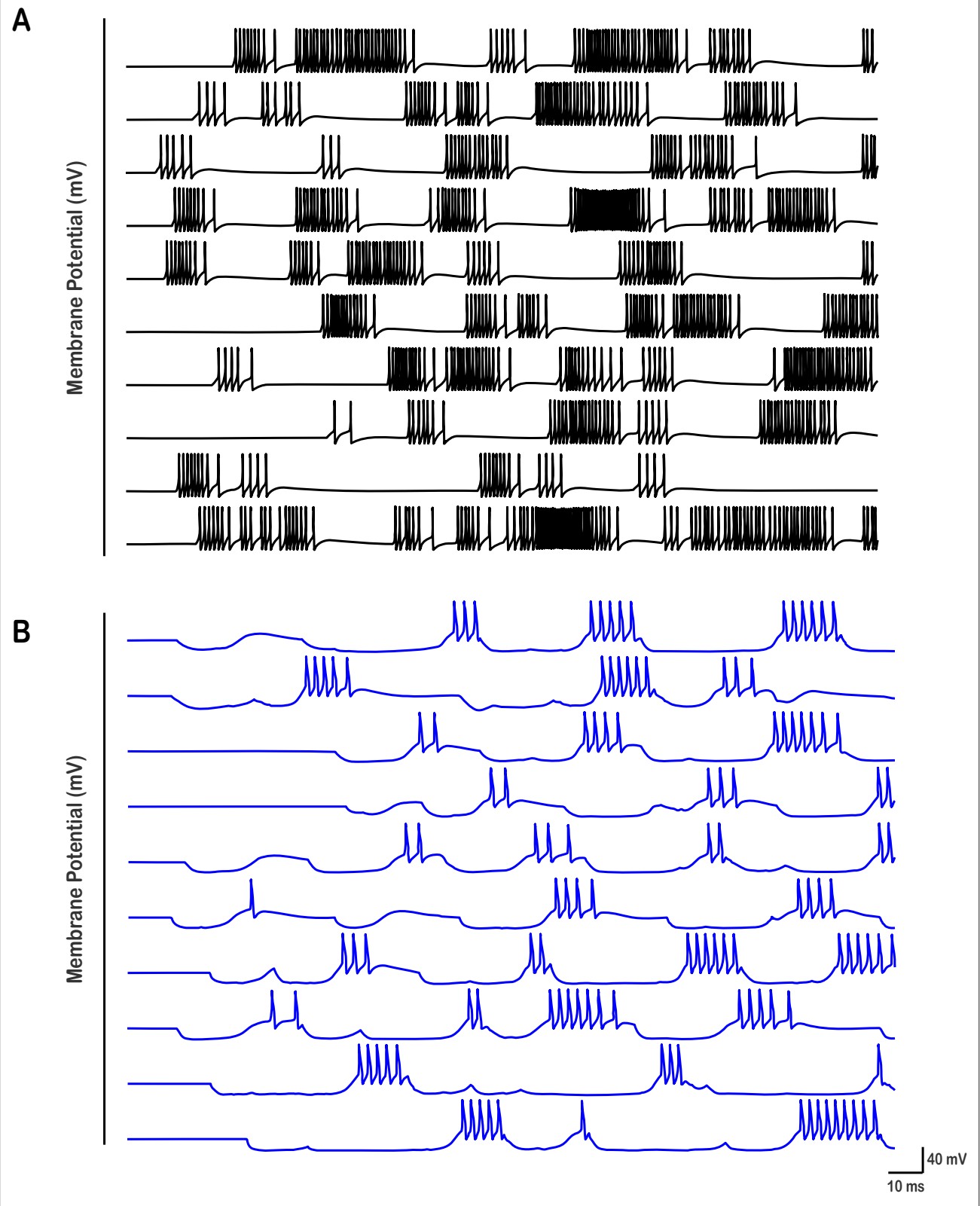

**Figure 7.** Activity patterns for 10 HVC$_{INT}$ and 10 HVC$_X$ neurons are illustrated. (**A**) HVC interneurons display dense spiking and bursting throughout the song, due to the dense HVC$_{RA}$ – HVC$_{INT}$ and HVC$_X$ – HVC$_{INT}$ excitatory coupling (*Figure 3*). (**B**) HVC$_X$ neurons display 2–4 rebound bursts that vary in their strength and duration due to HVC$_{INT}$ – HVC$_X$ inhibitory coupling as well as intrinsic properties (*Figure 8*).

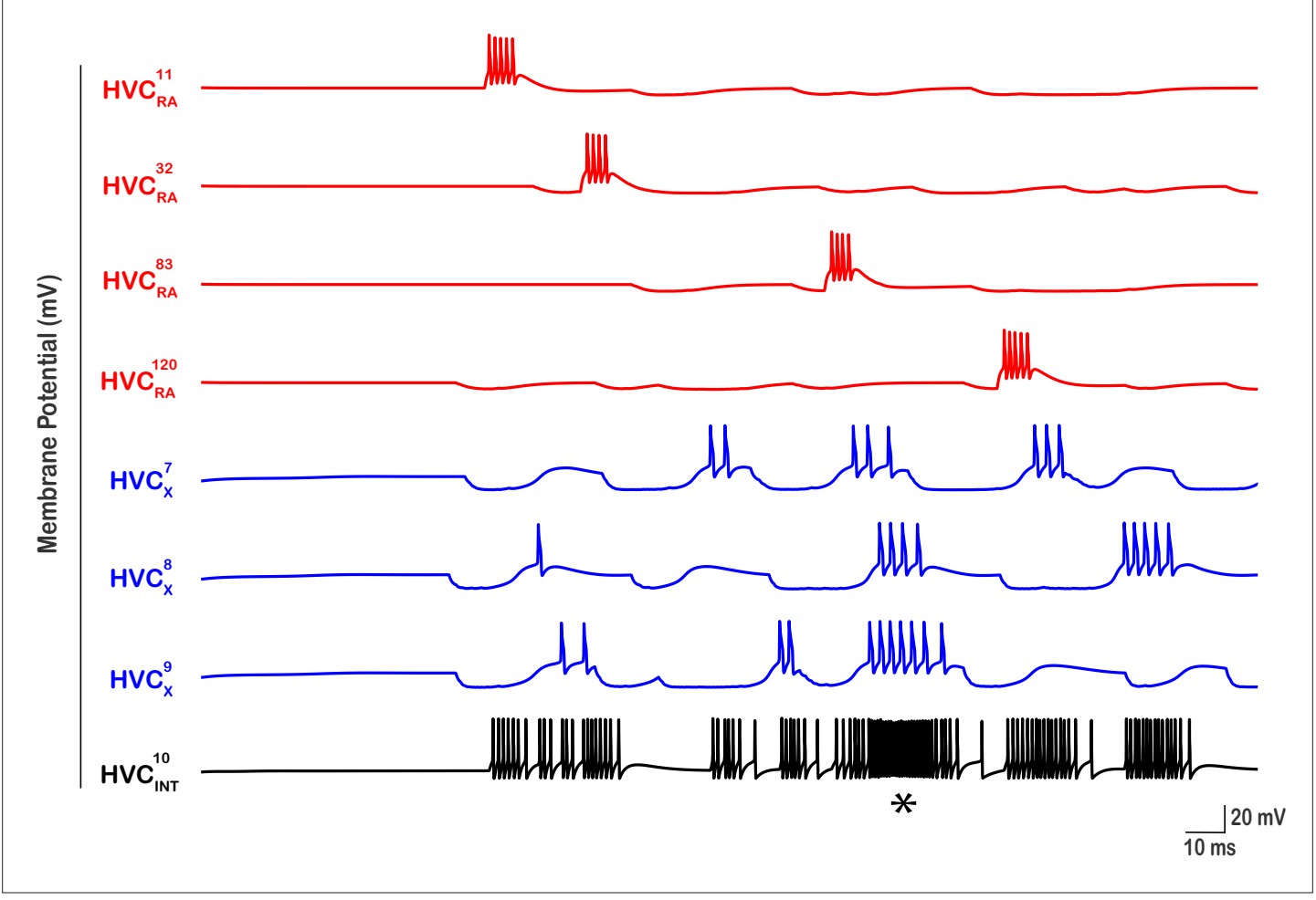

**Figure 8.** Patterned activity of HVC interneurons illustrated for an exemplar interneuron ($HVC_{INT}^{10}$). For this neuron, $HVC_{RA}^{11}$, $HVC_{RA}^{32}$, $HVC_{RA}^{83}$, $HVC_{RA}^{120}$, $HVC_{X}^{7}$, $HVC_{X}^{8}$ and $HVC_{X}^{9}$ were selected randomly from the pool of HVC_RA's and HVC_X's to form excitatory coupling. The number of bursts in $HVC_{INT}^{10}$ is controlled by the number of bursts that each of the HVC_RA and HVC_X neurons that connect to it exhibit. The strength of each of the $HVC_{INT}^{10}$ bursts depends on the magnitude of $g_{AMPA}$ from the corresponding neuron(s) they cause it as well as the simultaneous bursting of any of the projecting neurons. For example, the asterisk (*) shows a region of dense firing in $HVC_{INT}^{10}$ because $HVC_{RA}^{83}$, $HVC_{X}^{7}$, $HVC_{X}^{8}$ and $HVC_{X}^{9}$ neurons elicit their spikes at similar times causing a potentiated response in $HVC_{INT}^{10}$. HVC_X neurons exhibit multiple sags and rebounds because they're receiving inhibition from several interneurons (not shown here).

important feature of our model and is consistent with biological observations where neural populations exhibit robust collective behavior and the loss of a single neuron does not result in a major disruption of network activity.

## Bursting patterns of HVC_INT and HVC_X neurons

The activity patterns that HVC_INT neurons exhibit in our network architecture are illustrated in *Figure 7A* (shown here for 10 HVC_INT neurons). In contrast to the sparse activity in RA-projecting neurons, HVC interneurons generate multiple bursts and spike densely as reported during the song (*Hahnloser et al., 2002*). The number of bursts each HVC_INT neuron exhibits as well as the strength of each burst is controlled by network and synaptic mechanisms as described next (*Figures 8 and 9.*). HVC_X neurons, in their turn, generate 1–4 bursts per song motif (*Figure 7B*, shown here for 10 HVC_X neurons) similar to experimental results (*Fujimoto et al., 2011*; *Hahnloser et al., 2002*). Similarly, the number of bursts and strength of each HVC_X burst depends on a set of synaptic and intrinsic mechanisms, illustrated in *Figures 10 and 11*. HVC_X neurons differed in the number of bursts and the number of spikes per burst, rendering the results more realistic.

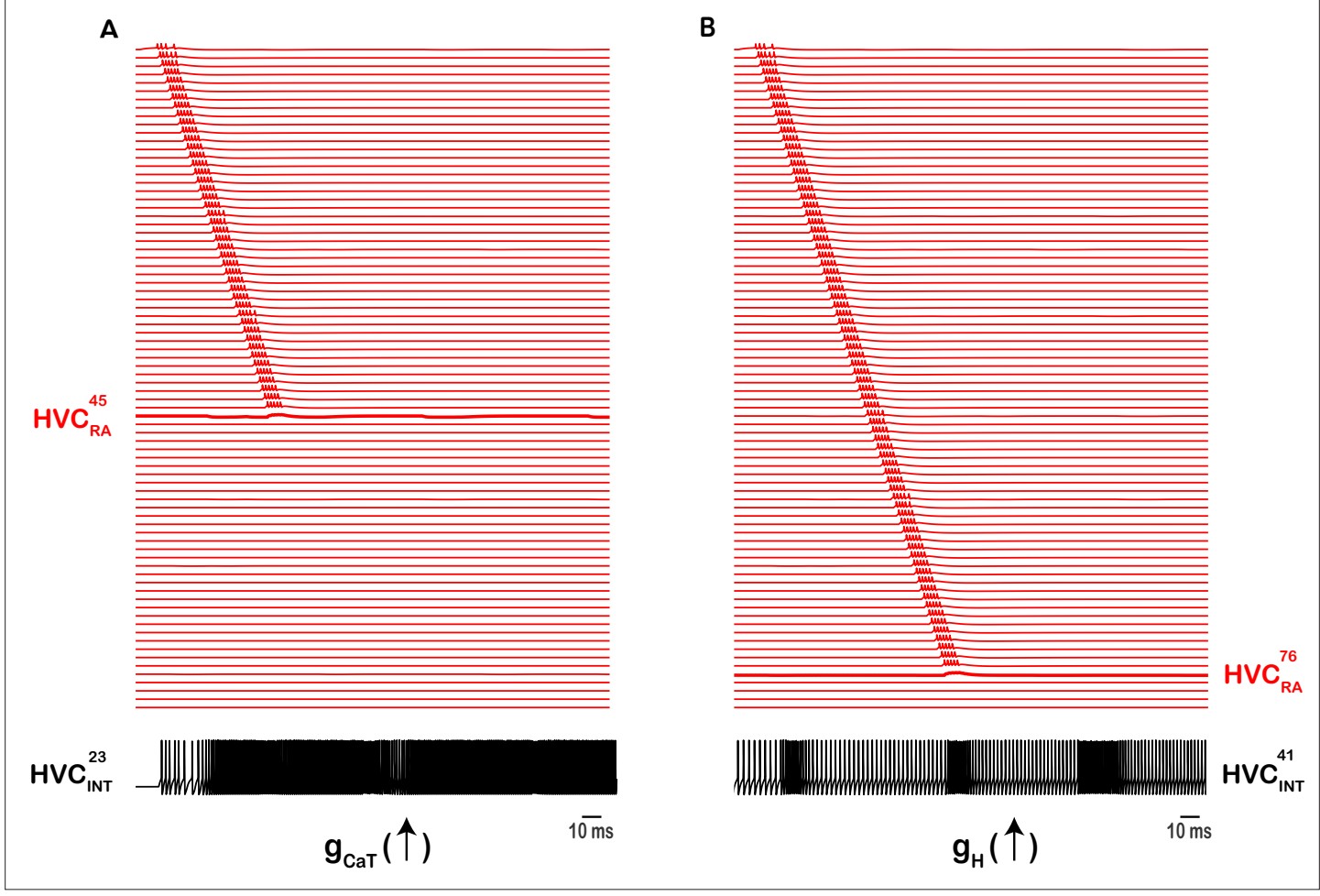

**Figure 9.** Intrinsic changes in HVC$_{INT}$ halt the propagation of sequential activity. Up-regulating the T-type Ca$^{2+}$ current conductance (**A**) or the hyperpolarization-activated inward current conductance (**B**) in exemplar HVC$_{INT}$ neurons eliminates sequence propagation. Increasing $g_{CaT}$ in $HVC_{INT}^{23}$ 10-fold results in dense bursting and firing in $HVC_{INT}^{23}$, which in its turn blocks the bursting of $HVC_{RA}^{45}$ due to the inhibitory GABA coupling between $HVC_{INT}^{23}$ and $HVC_{RA}^{45}$ (**A**). Similarly, increasing $g_H$ in $HVC_{INT}^{41}$ 10-fold results in increased firing in $HVC_{INT}^{41}$, which in its turn blocks the bursting of $HVC_{RA}^{76}$ due to the inhibitory GABA coupling between $HVC_{INT}^{41}$ and $HVC_{RA}^{76}$ (**B**). Sequence of HVC$_{RA}$ bursts truncated at the level of $HVC_{RA}^{80}$ for better visualization purposes.

## Dense bursting in HVC$_{INT}$

The random connections from HVC$_{RA}$ and HVC$_X$ neurons to HVC$_{INT}$ as well as the multiple bursts HVC$_{INT}$ and HVC$_X$ exhibit in this network are illustrated in *Figure 8* by highlighting a sample interneuron (HVC$_{INT}^{10}$). Here, the firing patterns of HVC$_{INT}^{10}$ in addition to all HVC$_{RA}$ and HVC$_X$ neurons that connect to it are shown. HVC$_{INT}^{10}$ received random synaptic inputs from HVC$_{RA}^{11}$, HVC$_{RA}^{32}$, HVC$_{RA}^{83}$ and HVC$_{RA}^{120}$ neurons, as well as from X-projecting neurons HVC$_X^7$, HVC$_X^8$ and HVC$_X^9$ in the same microcircuit it belongs to. Notice that each HVC$_{RA}$ neuron (which happens to belong to separate microcircuits in this case) bursts only once as reported experimentally (*Hahnloser et al., 2002*). HVC$_{INT}^{10}$ generates multiple bursts as well as spikes sparsely. Each of the HVC$_{INT}^{10}$ bursts is aligned with one of the bursts in HVC$_{RA}$ and/or HVC$_X$ neurons due to excitatory coupling.

The number of bursts an HVC$_{INT}$ neuron exhibits is largely determined by the number of bursts that each HVC$_{RA}$ and each HVC$_X$ neuron that synapses onto that interneuron exhibits. In other words, the larger the number of bursts that a population of HVC$_{RA}$ and HVC$_X$ that connects to an interneuron exhibits, the larger the number of bursts elicited in the interneuron. The strength of the bursts in HVC$_{INT}$ neurons is determined by two factors: (1) the strength of the excitatory synaptic conductance from HVC$_{RA}$ and/or HVC$_X$ to HVC$_{INT}$, with stronger bursts in HVC$_{INT}$ corresponding to larger magnitudes of $g_{AMPA}$ from HVC$_{RA}$ and/or HVC$_X$ onto the interneuron, (2) the number of spikes/bursts in

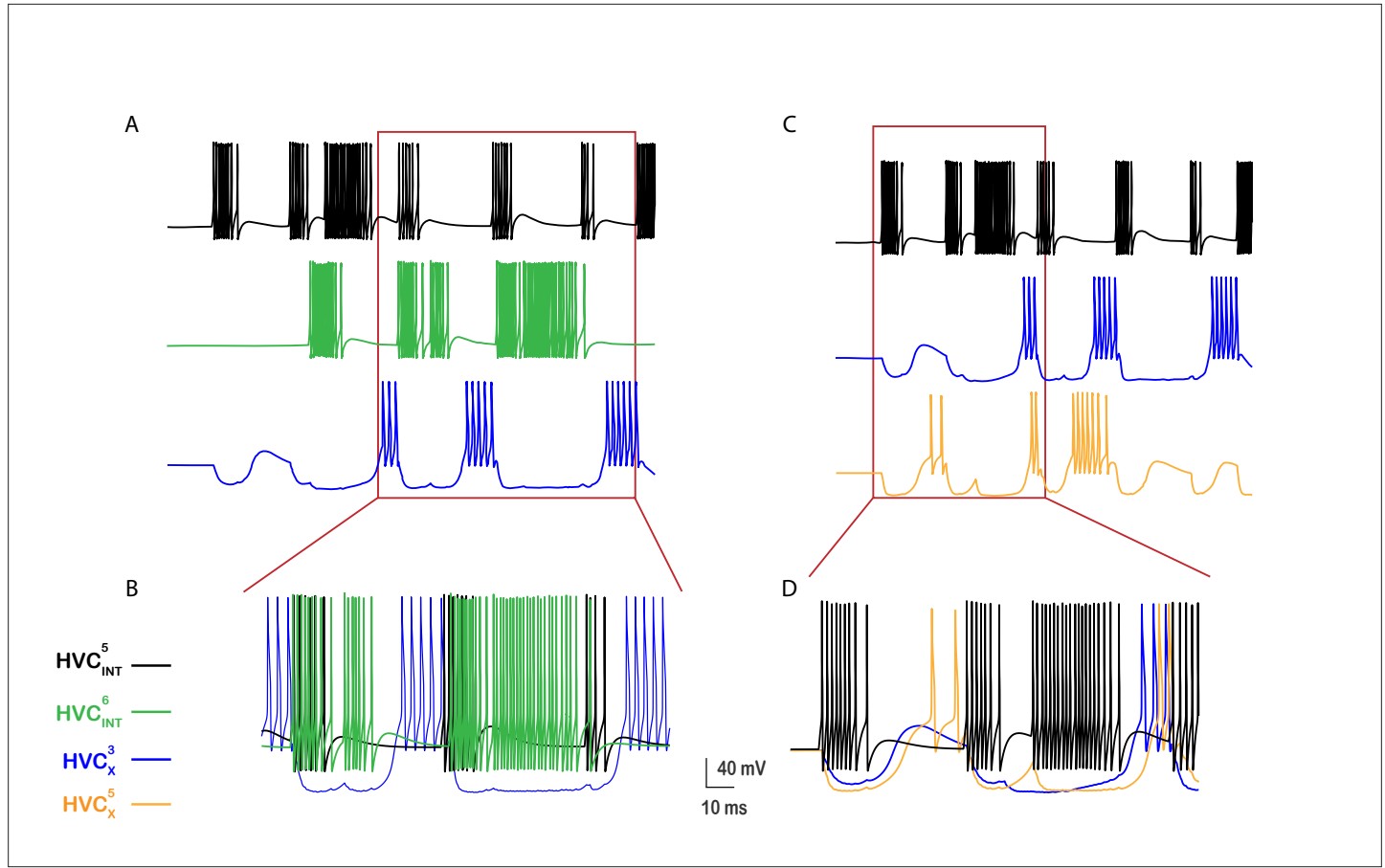

**Figure 10.** Activity patterns illustrating the interplay between HVC interneurons and X-projecting neurons. $HVC_X^3$ (blue) is an exemplar projecting neuron that receives inhibition from $HVC_{INT}^5$ (black) and $HVC_{INT}^6$ (green) due to the random inhibitory coupling (**A**). $HVC_X^3$ is inhibited whenever $HVC_{INT}^5$ and $HVC_{INT}^6$ are firing, eventually escaping inhibition at some intervals and eliciting rebound bursts due to the activation of $I_H$ and $I_{CaT}$. HVC interneurons can inhibit multiple HVC$_X$ neurons. $HVC_{INT}^5$ is an exemplar from the network that inhibits $HVC_X^3$ and $HVC_X^5$ (orange) (**C**). Bursts in $HVC_{INT}^5$ elicit subsequent bursts in $HVC_X^3$ and $HVC_X^5$ unless silenced by other HVC$_{INT}$ neurons that connect to them. Zoomed versions of (**A**) and (**C**) are shown in (**B**) and (**D**).

HVC$_{RA}$ and HVC$_X$ that are aligned and occur simultaneously at the HVC$_{INT}$ synapses. For example, the asterisk shown in *Figure 8* under the $HVC_{INT}^{10}$ spike train highlights a stronger burst (compared to the rest of the spiking pattern) because $HVC_{RA}^{83}$, $HVC_X^7$, $HVC_X^8$ and $HVC_X^9$ neurons tended to fire/burst at the same moment (or within a close interval of each other), thereby generating a stronger response in $HVC_{INT}^{10}$. Therefore, the characteristic tonic activity that HVC interneurons exhibit with bursting and suppression at different locations during singing is explained in our model by the excitatory and inhibitory coupling between the interneurons and both classes of projecting neurons (*Figure 12*).

The inhibitory effect of HVC$_{INT}$ neurons, their interplay with the excitatory projection neurons, and their intrinsic properties play a key role in the modulation of sequence propagation and overall desired network behavior. In particular, if the excitation from the projection neurons onto the interneurons was very large (>> g$_{AMPA}$), then HVC$_{INT}$ neurons enter regimes of very dense and continuous bursting/spiking (above their natural and basic, yet already enhanced potentiation), thereby leading to the inhibition of the HVC$_{RA}$'s, halting the sequence. Similarly, if the GABAergic conductances from HVC$_{INT}$ to HVC$_{RA}$ were relatively strong, outside their ideal ranges (*Figure 13B*), then HVC$_{RA}$ neurons won't be able to elicit their bursts.

Besides the synaptic modulations of HVC$_{INT}$ neurons on network activity, intrinsic mechanisms orchestrated primarily by the T-type Ca$^{2+}$ current and the hyperpolarization-activated inward current need to be tightly regulated to ensure desired network activity. The T-type calcium channel opens near resting membrane potential and markedly influences neuronal excitability (*Huguenard, 1996*;

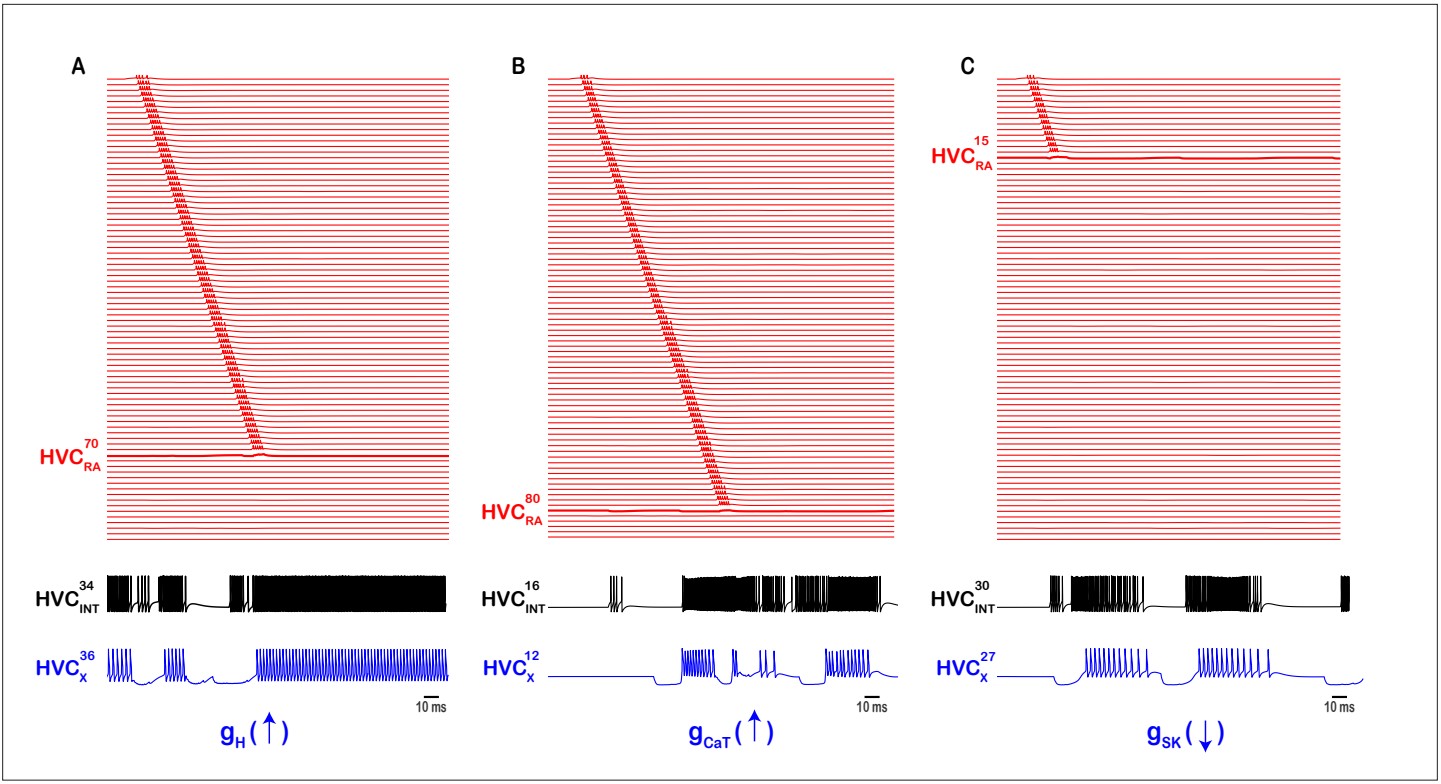

**Figure 11.** Intrinsic changes in HVC$_X$ halt the propagation of sequential activity. (**A**) Up-regulating the hyperpolarization-activated inward current conductance in a sample HVC$_X$ neuron ($HVC_x^{36}$, by increasing its $g_H$ 10-fold) leads to increased firing in all HVC$_{INT}$ neurons it connects to (for example, $HVC_{INT}^{34}$), which in its turn inhibits all HVC$_{RA}$ neurons it connects to (for example, $HVC_{RA}^{70}$, being first in the pool that it inhibits) breaking the sequence at the level of $HVC_{RA}^{70}$. (**B**) Up-regulating the T-type Ca2+ current conductance in a sample HVC$_X$ neuron ($HVC_x^{12}$, by increasing its $g_{CaT}$ 15-fold) leads to stronger rebound bursts in $HVC_x^{12}$ which leads to increased firing in all HVC$_{INT}$ neurons it connects to (e.g. $HVC_{INT}^{16}$), which in its turn inhibits all HVC$_{RA}$ neurons it connects to (for example, $HVC_{RA}^{80}$, being first in the pool that it inhibits) breaking the sequence at the level of $HVC_{RA}^{80}$. (**C**) Finally, down-regulating the Ca2+ - dependent K+ current conductance in a sample HVC$_X$ neuron ($HVC_x^{27}$, by setting its $g_{SK}$ to zero) leads to stronger rebound bursts in $HVC_x^{27}$ which leads to increased firing in all HVC$_{INT}$ neurons it connects to (e.g. $HVC_{INT}^{30}$), which in its turn inhibits all HVC$_{RA}$ neurons it connects to (for example, $HVC_{RA}^{15}$, being first in the pool that it inhibits) breaking the sequence at the level of $HVC_{RA}^{15}$. Sequence of HVC$_{RA}$ bursts truncated at the level of $HVC_{RA}^{85}$ for better visualization purposes.

*Jagodic et al., 2008*). In our model, if the T-type Ca$^{2+}$ current in an HVC$_{INT}$ neuron is up-regulated (due to a depolarizing shift in its voltage-dependent inactivation or simply setting $g_{CaT}$ to a relatively large value), then the interneuron will fire with much larger firing frequency, silencing the corresponding HVC$_{RA}$ and HVC$_X$ neurons that it connects to and breaking the sequence (*Figure 9A*). Similarly, the H-channels regulate the resting membrane potentials of the neurons they're expressed in and play a key role in regulating the spontaneous firing activity (*Datunashvili et al., 2018*; *Funahashi et al., 2003*; *Yao et al., 2003*). In our model, if the H conductance of an HVC$_{INT}$ was upregulated (by increasing the magnitude of $g_h$), then the neuron switches to continuous firing, silencing the HVC$_{RA}$'s that it connects to *Figure 9B*. Therefore, both the interplay between excitation and inhibition between HVC$_{INT}$ and HVC projection neurons as well as the intrinsic properties of HVC$_{INT}$ neurons ($I_{CaT}$ and $I_H$) are necessary elements to ensure an accurate propagation of sequential activity.

## Rebound bursting in HVC$_X$

The activity patterns that HVC$_X$ neurons exhibit in the network are illustrated in *Figure 7B* (shown here for 10 HVC$_X$ neurons). The X-projecting neuronal firing is characterized by regions of inhibition throughout singing, interrupted by occasional rebound bursting. The strength of each burst and the number of total bursts in an HVC$_X$ depend on synaptic and intrinsic mechanisms summarized briefly as such: (1) the degree of inhibition from HVC$_{INT}$ neurons, (2) the number and timing of bursts of HVC$_{INT}$ neurons and (3) the intrinsic properties and magnitudes of the T - type Ca$^{2+}$ and H - currents that the HVC$_X$ neurons exhibit. In short, the stronger the GABAergic maximal conductance(s) from

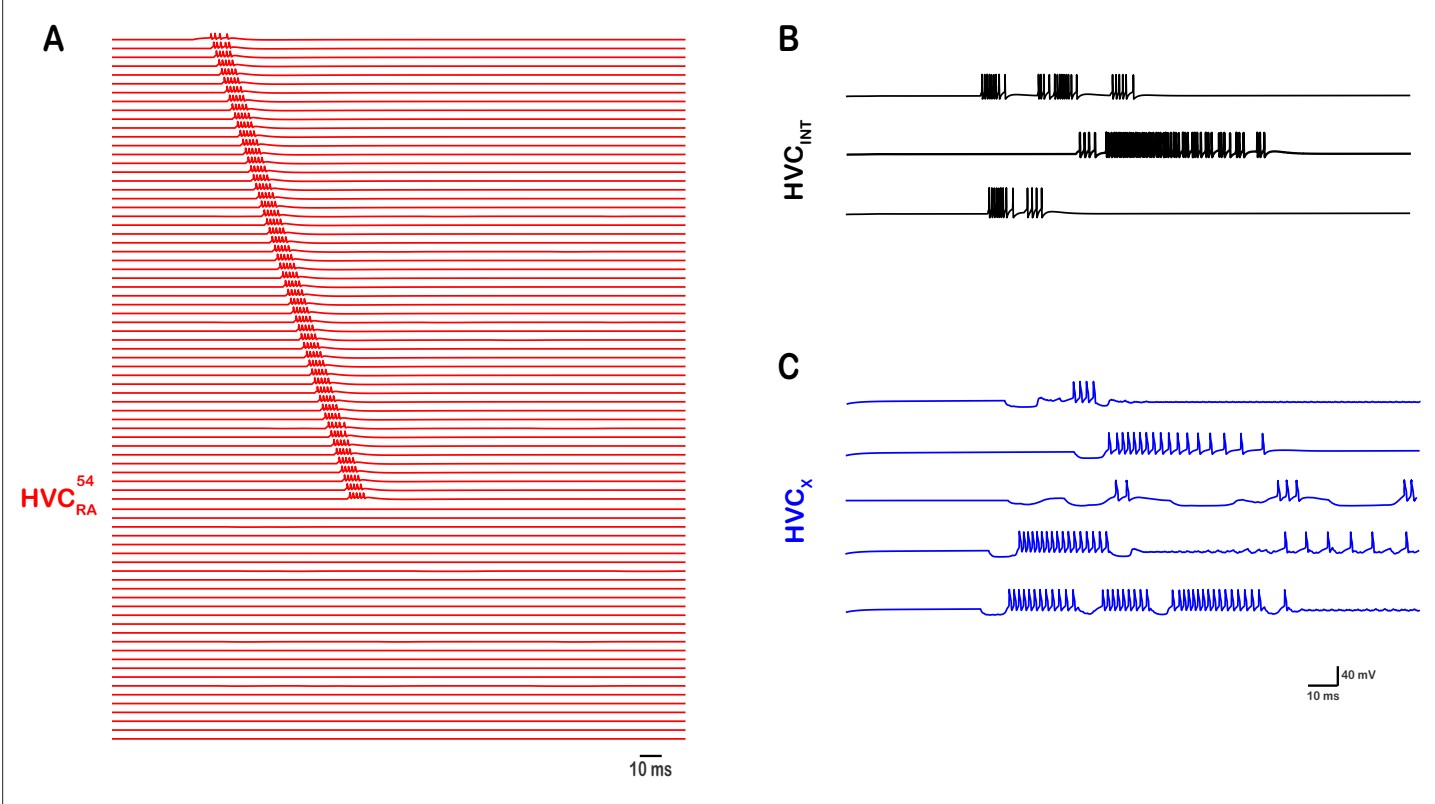

**Figure 12.** Altering the intrinsic properties of HVCx neurons disrupts network activity in 59% of the cases (out of 100 simulations) where the maximal conductances (gCaT, gSK, and gH) of HVCx neurons are randomly varied within their allowed ranges (*Figure 13A*). As a result, some HVC interneurons (**B**) and X-projecting neurons (**C**) generated biologically unrealistic firing patterns, halting the propagation of sequential activity in RA-projectors (**A**).

the $HVC_{INT}$ neurons that inhibit a given $HVC_X$ neuron, the stronger the $HVC_X$ neuron's rebound burst (*Figures 7–10*). Similarly, if multiple $HVC_{INT}$ neurons' bursts that inhibit an $HVC_X$ neuron aligned simultaneously, then the rebound in $HVC_X$ is potentiated due to the stronger inhibition. This is primarily due to the T-type $Ca^{2+}$ current and the H-current that $HVC_X$ neurons exhibit, facilitating rapid calcium influx into the neurons when they rebound from hyperpolarization. The calcium influx is correlated to the degree and the duration of inhibition that the neuron receives and can trigger a potentiated burst of action potentials leading to more robust rebound responses. In other words, the stronger the inhibition of $HVC_X$, the stronger the activation of $I_{CaT}$ and $I_h$, leading to stronger rebounds. Moreover, the number of spikes in each $HVC_X$ burst is controlled by several factors including (1) the degree of inhibition from $HVC_{INT}$ and its effect on $I_{CaT}$ and $I_h$ as described earlier, and (2) the strength of the $Ca^{2+}$-dependent $K^+$ conductance ($g_{SK}$) with stronger magnitudes of this conductance dampening the excitability of these neurons and reducing the number of spikes.

The interplay between $HVC_{INT}$ and $HVC_X$ in shaping the characteristic $HVC_X$ responses is illustrated in *Figure 10* from exemplar neurons in the network. Two interneurons, $HVC_{INT}^5$ (black) and $HVC_{INT}^6$ (green), were selected randomly to inhibit $HVC_X^3$ (blue trace, *Figure 10A*). There are several regions in the $HVC_X^3$ membrane potential trajectory where the neuron is inhibited (sags/sinks in the voltage trace). These regions correspond to the moments in time when either $HVC_{INT}^5$ or $HVC_{INT}^6$ neurons, or both simultaneously, are bursting, thereby silencing $HVC_X^3$. Eventually, $HVC_X^3$ is able to escape inhibition at a few instances in time and generate multiple post-inhibitory rebound bursts mediated by $I_{CaT}$ and $I_h$. *Figure 10B* shows a zoomed version of panel A illustrating the three rebound bursts in $HVC_X^1$ as a result of escape from inhibition. $HVC_{INT}^5$ and $HVC_{INT}^6$ neurons generated multiple successive bursts of firing, during which $HVC_X^3$ cannot escape the inhibition. There are only a few intervals where $HVC_X^3$ is able to elicit rebound bursts; the first opportunity was at the beginning of the trace (*Figure 10A*) where the sag generated in $HVC_X^3$ as a result of inhibition from $HVC_{INT}^5$ is clear, but the inhibition from $HVC_{INT}^5$ was not strong enough to elicit a rebound burst in $HVC_X^3$, rather eliciting only

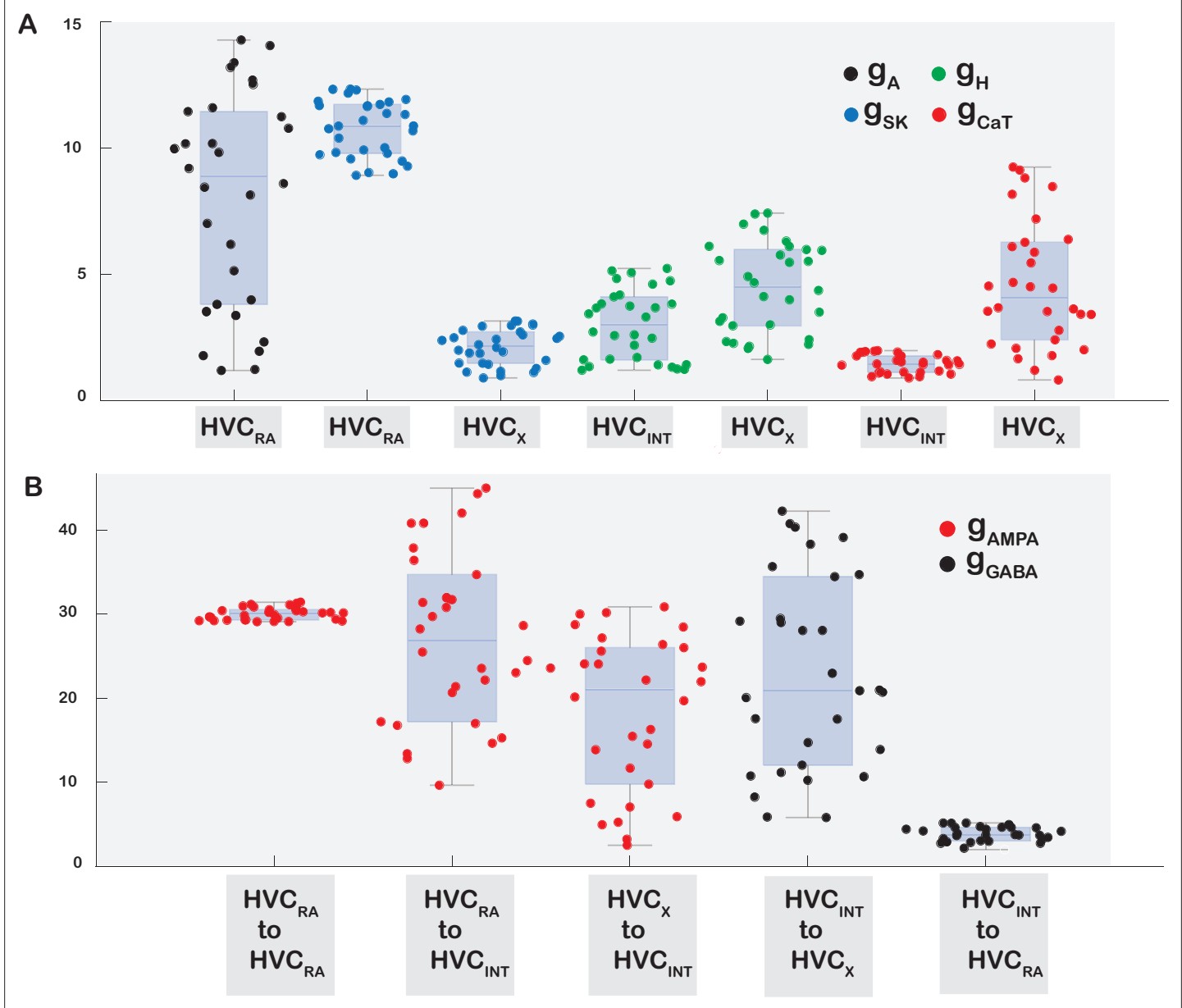

**Figure 13.** Box plots showing the ranges of ionic and synaptic currents that were allowed to vary while maintaining robust network propagation and biologically realistic in vivo behavior of all HVC neuronal classes. (**A**) The ionic conductances that were varied are $g_A$ of $HVC_{RA}$, $g_{SK}$ of $HVC_{RA}$ and $HVC_X$, $g_h$ and $g_{CaT}$ of $HVC_{INT}$ and $HVC_X$. The shown ranges reflect values whereby each neuron class is able to maintain realism in terms of electrophysiological behavior and network properties. (**B**) Ranges of values of synaptic conductances that connect two classes of HVC neurons while conserving sequential activity propagation and the general network activity.

a rebound depolarization. The subsequent three opportunities where $HVC_X^3$ is able to escape the inhibition all elicited rebound bursts due to the strong inhibitory input arriving from both interneurons. All rebound bursts in $HVC_X$ are orchestrated by the hyperpolarization-activated inward current ($I_H$) and the T-type $Ca^{2+}$ current ($I_{CaT}$). The longer the duration of the inhibition prior to a rebound, the longer the time $I_H$ and $I_{CaT}$ take to fully activate and generate the rebound burst. The shorter the inhibitory silent interval (i.e. the interval in which $HVC_X$ is receiving no inhibition), the shorter the rebound and the fewer the number of spikes in the rebound burst (*Figure 10A*, first rebound burst). Moreover, HVC interneurons inhibit multiple $HVC_X$ neurons in our network. *Figure 10C* shows an exemplar interneuron ($HVC_{INT}^5$, black) that happens to inhibit two X-projecting neurons ($HVC_X^3$, blue and $HVC_X^5$, orange). Each burst in $HVC_{INT}^5$ elicits an iPSP in $HVC_X^3$ and $HVC_X^5$, and can contribute to

subsequent rebound bursts in them unless silenced by other interneurons that connect to $HVC_X^3$ or $HVC_X^5$ (*Figure 10D*). All $HVC_X$ neurons differed in their corresponding number of rebound bursts as well as in the number of spikes per burst, rendering the results biologically accurate.

As mentioned earlier, the T-type $Ca^{2+}$ and the H- conductances play a significant role in modulating the rebound bursting in $HVC_X$ neurons. Due to this interplay, we do not need significant inhibition to generate rebound bursts, because the T-type $Ca^{2+}$ current's conductance can be stronger than the inhibitory conductance, leading to robust rebound bursting, even when the degree of inhibition is not very strong. As a matter of fact, these two conductances, along with the $Ca^{2+}$ -dependent $K^+$ conductance, can halt the sequential propagation of activity and mess up the overall desired network behavior as described next. Ramping up $g_H$ in $HVC_X$ neurons to values outside the allowed ranges (*Figure 13A*) will break the sequence of propagation and generate nonrealistic firing patterns. The $g_H$ parameter was increased in *Figure 11A* in an exemplar $HVC_X$ neuron ($HVC_X^{36}$) to large values, driving $HVC_X^{36}$ into regimes of runaway excitation due to the up-regulation of $I_H$ and generating a non-realistic firing pattern for a typical X-projecting neuron during singing. The reason the sequential propagation of $HVC_{RA}$ is halted because $HVC_X^{36}$ happens to have an excitatory coupling with $HVC_{INT}^{34}$ (along with other interneurons, but we illustrate it here for $HVC_{INT}^{34}$), and as a result of $HVC_X^{36}$'s increased firing, $HVC_{INT}^{34}$ generates a mostly continuous firing trace of bursting and spiking, which silences the $HVC_{RA}$ neuron that it happens to inhibit (in this case, $HVC_{RA}^{70}$, *Figure 11A*).

Similarly, altering $g_{CaT}$ in $HVC_X$ neurons can have similar effects on network activity. *Figure 11B* shows an exemplar $HVC_X$ neuron ($HVC_X^{12}$) where $g_{CaT}$ was ramped up to large values and as a result, stronger rebound bursts in $HVC_X^{12}$ were elicited as well as larger number of rebound spikes in each burst. This is primarily due to the up-regulation of the T-type $Ca^{2+}$ current that was induced, markedly influencing the neuronal excitability. The stronger rebounding in $HVC_X^{12}$ generated stronger excitation in their postsynaptic interneurons' counterparts that they excite (e.g. $HVC_{INT}^{16}$), which in their turn silenced the $HVC_{RA}$ neurons that they inhibit (for example, $HVC_{RA}^{80}$ is the first neuron in the chain that $HVC_{INT}^{16}$ inhibits, breaking the sequence at $HVC_{RA}^{80}$). And finally, the $Ca^{2+}$ -dependent $K^+$ conductance, which plays a key role in governing $HVC_X$ neurons' excitability and their characteristic spike frequency adaptation (*Daou et al., 2013*), can have similar effects if the channel was down-regulated (rather than up-regulated as in $I_{CaT}$ and $I_H$). *Figure 11C* shows the effects of reducing the $g_{SK}$ conductance in an exemplar $HVC_X$ neuron ($HVC_X^{27}$) leading to its increased firing, which in turn leads to stronger bursting and spiking in the $HVC_{INT}$ neurons it excites (e.g. $HVC_{INT}^{30}$). $HVC_{INT}^{30}$ in its turn generates stronger and wider-range inhibition onto the $HVC_{RA}$ neurons it sends its axons to (e.g. $HVC_{RA}^{15}$ is the first neuron in the chain $HVC_{INT}^{30}$ inhibits, silencing it and breaking the sequence at $HVC_{RA}^{15}$).

Finally, we checked the consequences of altering the intrinsic properties of $HVC_X$ neurons on the network's desired behavior. To do so, we varied the maximal conductances of the three principal ionic currents of the X-projecting neurons ($I_{CaT}$, $I_{SK}$, $I_H$) across all neurons of the population, while keeping this variation within the reported ranges shown in *Figure 13A* (because certainly going outside these ranges will disrupt network activity for other reasons as reported in *Figures 9 and 11*). Varying those three key parameters across the HVCx population had different results. In 100 different simulations that generated random maximal conductances for $I_{CaT}$, $I_{SK}$, and $I_H$, 41% of the simulations did not have any considerable effect on the desired network activity, whereas 59% resulted in disrupted network activity, and sometimes detrimental. For example, *Figure 12* shows an example where the sequential propagation of activity was halted and the firing patterns of some interneurons and X-projecting neurons were rendered non-biophysically realistic. Particularly, some interneurons switched to continuous spiking or phasic bursting with little episodic bursting modes, while some $HVC_X$ neurons generated longer rebounds, fewer number of bursts, or fewer number of spikes per burst (*Figure 12B–C*). Hence, changes in the intrinsic properties of X-projecting neurons can disrupt activity propagation necessary for song production and produce biologically unrealistic bursting patterns in HVC neurons. This can wreak havoc on our network model hinting to the finding that biophysical parameters are distinct and consistent for an individual bird and this unique combination is needed for song (*Daou and Margoliash, 2020*). The homogeneity in the intrinsic properties of X-projectors might be a strategy allowing it to adapt or respond to changes in the network.

In conclusion, we developed a detailed and biophysically realistic neural network model for sequence propagation in the HVC of the zebra finch. Our model consisted of chains of microcircuits, each comprised of a selection of $HVC_{RA}$, $HVC_{INT}$, and $HVC_X$ model neurons selected randomly from a

total pool of neurons. The maximal conductances of the four key and principal ionic currents for each model neuron, the number of neurons of each class in any microcircuit, and the excitatory and inhibitory connections between the different classes within a microcircuit and across microcircuits are all selected randomly. This activity propagates throughout the chain of microcircuits causing a sequence of HVC$_{RA}$ bursts while leaving behind realistic bursting patterns for all classes of HVC neurons as seen during singing. The model incorporates all known ionic and synaptic currents for each HVC neuron. The network architecture we developed was able to replicate the in vivo biologically realistic firing behavior for each class by including sparse timely-locked bursting in the RA-projecting neurons (with accurate intrinsic properties for each burst in terms of number of spikes, duration, and spike morphology), multiple bursting in the X-projecting neurons that are also sparse and time-locked, and dense bursting/spiking in the interneurons with few intermittent quiescence. The ability of our network to reproduce the sequential propagation of activity in the presence of excitatory and inhibitory connections involving all neuronal subclasses as well as over a range of values for each synaptic and ionic maximal conductances is an indication of its robustness. Our network unveiled key intrinsic and synaptic mechanisms that modulate the sequential propagation of neural activity by highlighting important roles for the T-type Ca$^{2+}$ current and hyperpolarization-activated (H) inward current in HVC$_X$ and HVC$_{INT}$ neurons, Ca$^{2+}$-dependent K$^+$ current in HVC$_X$ and HVC$_{RA}$, A-type K$^+$ current in HVC$_{RA}$, as well as GABAergic and glutamatergic synaptic currents that connect all neuronal subclasses together. The result is an improved characterization of the HVC network responsible for song production in the zebra finch. Beyond replicating established HVC firing patterns, our model provides testable hypotheses that intrinsic membrane properties, particularly inhibitory timing and rebound bursts, maintain robust sequential propagation. This context generates clear experimental predictions: for example, modulating I$_H$ or I$_{CaT}$ in HVC$_X$ or HVC$_{INT}$ neurons directly affects rebound spiking and sequence propagation; or altering I$_{SK}$ or I$_A$ conductances disrupts the ability of HVC$_{RA}$ neurons to burst at the right time. The model also suggests that structured inhibition can act as a temporal scaffold for burst timing guiding experiments that manipulate interneuron dynamics or the overall network inhibition through optogenetics or pharmacology. Hence, a significant strength of the proposed model is that it puts forward suggestions for experimental manipulations in the form of targeted experiments whether using optogenetics or pharmacology that would help validate the model's mechanisms and further clarify the specific roles different HVC components play in driving sequential activity.

## Discussion

In this study, we developed a biophysically realistic neural network model to explore how intrinsic neuronal properties and local connectivity within the songbird nucleus HVC may support the generation of temporally precise activity sequences associated with zebra finch song. The biophysically realistic network architecture that we designed combines both classes of HVC projection neurons with local inhibitory interneurons. A fundamental goal that we have achieved in our design is a successful replication of the in vivo firing behaviors of all the HVC neuronal classes: single sparse timely-precise bursting (3–6 spikes for ~10ms) in the RA-projecting neurons, multiple bursting (1–4 bursts with 4–9 spikes/burst) in the X-projecting neurons, dense and frequent bursting in the interneurons, as well as the general intrinsic properties that each class of HVC neurons exhibit (*Daou et al., 2013*; *Lewicki, 1996*; *Long et al., 2010*). The patterning activity in HVC is largely shaped in our model by the intrinsic properties of the individual neurons as well as the synaptic properties where excitation and inhibition play a major role in enabling neurons to generate their characteristic bursts during singing.

The three classes of model neurons incorporated to our network as well as the synaptic currents that connect them are based on Hodgkin-Huxley formalisms that contain ion channels and synaptic currents which had been pharmacologically identified (*Daou et al., 2013*; *Kosche et al., 2015*; *Mooney and Prather, 2005*). Our network showed that sequence propagation can be broken if several intrinsic mechanisms are perturbed. In particular, if I$_{CaT}$ or I$_H$ is upregulated in HVC$_X$ or HVC$_{INT}$, if I$_{SK}$ is downregulated in HVC$_X$ or if I$_{SK}$ is upregulated in HVC$_{RA}$, then the corresponding chain of activity stops and the rhythmic activity of the network is disrupted (*Figures 6, 9 and 11*). Synaptically, perhaps the most critical role in our network design is played by interneurons which orchestrate the activity of the two projection neurons in a structured manner. Interneurons adjust the timing of HVC projection neurons' bursts (*Amador et al., 2013*; *Kosche et al., 2015*), and developmental learning regulates inhibition onto HVC$_{RA}$ (*Vallentin et al., 2016*). While findings of *Kosche et al., 2015* emphasize the robustness

of the HVC timing circuit to inhibition, our model is more sensitive to inhibition, highlighting that HVC likely operates with several redundant mechanisms that overall ensure temporal precision. $HVC_{RA}$ neurons interact with $HVC_X$ through local interneurons, a disynaptic inhibitory pathway that conveys information to $HVC_X$ neurons (*Prather et al., 2008*). Focal application of the $GABA_A$ receptor antagonist, gabazine, restricted inhibitory impact in HVC leading to stronger and faster responses relative to call onset (*Benichov and Vallentin, 2020*), showing that local HVC interneurons form an inhibitory mask that can greatly constrain the spiking activity of projecting neurons (*Kornfeld et al., 2017*; *Kosche et al., 2015*; *Mooney and Prather, 2005*), suggesting that HVC model networks that lack inhibitory neurons are inadequate for explaining sequential propagation of neural activity.

Various models of how the song is encoded within HVC have been proposed. Some groups suggested that bursting activity propagates through a chain of synaptically connected $HVC_{RA}$ neurons either as single neurons (*Fee et al., 2004*; *Hahnloser et al., 2002*; *Long et al., 2010*) or as pools of $HVC_{RA}$ neurons, each group driving a distinct ensemble of RA neurons (*Jin et al., 2007*; *Leonardo and Fee, 2005*; *Li and Greenside, 2006*). These models assume that $HVC_{RA}$ neurons generate a continuous, feed-forward sequence of activity over time, with little or no role played by X-projecting HVC neurons and interneurons. Other models have incorporated alternative temporal encoding mechanisms by necessitating synaptic integration at the levels of $HVC_{RA}$ and $HVC_{INT}$ populations (*Drew and Abbott, 2003*; *Gibb et al., 2009a*; *Jin, 2009*; *Weber and Hahnloser, 2007*) while yet other approaches gave emphasis to brainstem feedback processes by incorporating inter-hemispheric coordination to activate sequences of syllable-specific $HVC_{RA}$ and $HVC_{INT}$ neurons (*Galvis et al., 2018*; *Gibb et al., 2009b*). A prominent model used spatially recurrent excitatory chains and local feedback inhibition to show how the HVC network stabilizes synchrony while propagating sequential activity (*Cannon et al., 2015*; *Markowitz et al., 2015*).

All existing models that describe premotor sequence generation in the HVC either assume a distributed model (*Elmaleh et al., 2021*) that dictates that local HVC circuitry is not sufficient to advance the sequence but rather depends upon moment-to-moment feedback through Uva (*Hamaguchi et al., 2016*), or assume models that rely on intrinsic connections within HVC to propagate sequential activity. In the latter case, some models assume that HVC is composed of multiple discrete subnetworks that encode individual song elements (*Glaze and Troyer, 2013*; *Long and Fee, 2008*; *Wang et al., 2008*), but lacks the local connectivity to link the subnetworks, while other models assume that HVC may have sufficient information in its intrinsic connections to form a single continuous network sequence (*Long et al., 2010*).

The network architecture we developed here exhibits overlap with the various models presented. First, in agreement with the continuous model, our network architecture displays a feed-forward mechanism regulating the circuit dynamics (e.g. *Gibb et al., 2009a*; *Jin, 2009*; *Jin et al., 2007*; *Li and Greenside, 2006*; *Long et al., 2010*). Nonetheless, diverging from a linear progression of HVC neurons directing the song, the network's structure comprises sequences of microcircuits incorporating all classes of HVC neurons, where sequential activity transmits from one microcircuit to the next, as opposed to transitioning directly between individual neurons. Second, in agreement with the subnetwork models, our model envisions HVC as comprised of multiple discrete subcircuits (SSSs) where each microcircuit incorporates its own pool of neurons; however, in our model HVC's connectivity is sufficient to link the microcircuits together and extrinsic influences are not needed. Moreover, our network is in agreement with the (*Cannon et al., 2015*) model where structured inhibition is needed to propagate sequential activity, synchronize the firing of pools of neurons, and stabilize spike timing along the chain. The pivotal element in advancing sequential activity through time is the inhibition exerted by $HVC_{INT}$ onto $HVC_X$ and $HVC_{RA}$ neurons, facilitated in the case of $HVC_X$ through rebound firing, and all orchestrated by intrinsic mechanisms.

A potential drawback of our model is that it does not incorporate brainstem feedback processes or address inter-hemispheric coordination as proposed by others (*Galvis et al., 2018*; *Gibb et al., 2009b*). Another drawback is its sole focus on local excitatory connectivity within the HVC (*Kornfeld et al., 2017*; *Long et al., 2010*). Moreover, HVC neurons receive afferent excitatory connections (*Akutagawa and Konishi, 2010*; *Nottebohm et al., 1982*) that play significant roles in their local dynamics. For example, the excitatory inputs that HVC neurons receive from Uvaeformis may be crucial in initiating (*Andalman et al., 2011*; *Danish et al., 2017*; *Galvis et al., 2018*) or sustaining (*Hamaguchi et al., 2016*) the sequential activity. In addition, while our simplified, somatically driven

architecture enables better exploration of mechanisms for sequence propagation, future extensions of the model will incorporate dendritic compartments to more accurately reflect the intrinsic bursting mechanisms observed in HVC$_{RA}$ neurons. Moreover, our model was run at a fixed physiological temperature, but it is well known going all the way back to Hodgkin and Huxley that both ion channel activity and synaptic dynamics can change with temperature. In future work, adding temperature scaling (like Q10 factors) could help us explore how burst timing and sequence speed change with temperature changes, and how neural activity in HVC would/would not preserve its precision under different physiological conditions.

### The role of ion channels in controlling network activity

Our model highlights the role of principal ion channels (I$_H$, I$_{CaT}$, I$_{SK}$, and I$_A$) in controlling HVC's network dynamics and progressing its neural sequence. Hyperpolarization-activated ionic conductances had been widely observed across various electrically excitable cells (*Pape, 1996*) and play significant roles in rhythmogenesis (*Budde et al., 1997*; *Golowasch et al., 1992*; *Golowasch and Marder, 1992*). In our network, model HVC$_X$ neurons are not able to elicit their rebound bursting without I$_H$ and sequence is halted if this conductance is upregulated in either of HVC$_X$ or HVC$_{INT}$ (*Figures 7–9, 10 and 11*). Similarly, the T-type Ca$^2$+current is recognized as crucial in various systems as an ionic contributor to burst generation (*Deschênes et al., 1982*; *Fraser and MacVicar, 1991*; *Huguenard, 1996*; *Llinás and Yarom, 1981*). *Lewicki, 1996* observed a significant hyperpolarization in some HVC neurons in vivo before they emit their corresponding bursts, and the intensity of the burst correlates with the degree of hyperpolarization. In this study, we have illustrated its pivotal role in rebound spiking where up-regulating this conductance in HVC$_X$ or HVC$_{INT}$ halts sequence propagation (*Figures 7–9, 10 and 11*). Moreover, the A-type K$^+$ current is involved in several rhythmogenic activities controlling membrane excitability (*Coetzee et al., 1999*; *Ellis et al., 2007*; *Gross et al., 2016*) and in our network, upregulating I$_A$ suppress bursting in model HVC$_{RA}$ and breaks sequence propagation (*Figure 6*). Finally, the small conductance Ca$^{2+}$-activated potassium current (I$_{SK}$) plays important roles in the regulation of excitable cells controlling network rhythmic activity (*Benítez et al., 2011*; *Chen et al., 2014*; *Pedarzani et al., 2005*) and in our network I$_{SK}$ plays a significant role since its upregulation in HVC$_{RA}$ or its downregulation in HVC$_X$ eliminates sequence propagation (*Figures 6 and 11* respectively).

In conclusion, the network model developed provides a large step forward in describing the biophysics of HVC circuitry and may throw a new light on certain dynamics in the mammalian brain, particularly the motor cortex (*Shmiel et al., 2006*) and the hippocampus regions (*Lee and Wilson, 2002*) where precisely timed sequential activity is crucial. We suggest that temporally precise sequential activity may be a manifestation of neural networks comprised of chains of microcircuits, each containing pools of excitatory and inhibitory neurons, with local interplay among neurons of the same microcircuit and global interplays across the various microcircuits, and with structured inhibition and intrinsic properties synchronizing the neuronal pools and stabilizing timing within the ongoing sequence.

## Materials and methods

Single-compartment conductance-based Hodgkin-Huxley-type (HH) biophysical models of cells from the HVC were developed and connected together via biologically realistic synaptic currents. Simulations of these model neurons and of the model network composed of synaptically coupled HVC$_{RA}$, HVC$_X$, and HVC$_{INT}$ neurons were performed using the ode45 numerical integrator in MATLAB (Math-Works). Source codes for each network will be made available online at our lab's website as well as on ModelDB.

HVC model cells that are used to connect the networks exhibited ionic and synaptic currents that had been shown to be expressed pharmacologically (*Daou et al., 2013*; *Kosche et al., 2015*; *Long et al., 2010*; *Mooney and Prather, 2005*). The functional forms of activation/inactivation functions and time constants were based on previous published mathematical neural models (*Daou et al., 2013*; *Destexhe and Babloyantz, 1993*; *Dunmyre et al., 2011*; *Hodgkin and Huxley, 1952*; *Terman et al., 2002*; *Wang et al., 2003*), and the parameters that were varied were merely the maximal conductances of some ionic currents that vary among the various neuronal subtypes (*Daou et al., 2013*), as

well as the synaptic conductances. Every model neuron is represented by ordinary differential equations for the different state variables as illustrated below.

## Ion channels of model HVC neurons

We added a hyperpolarization-activated inward current conductance ($I_H$) to $HVC_X$ and $HVC_{INT}$ because it is responsible for the sag seen in these neurons (*Daou et al., 2013*; *Dutar et al., 1998*; *Kubota and Saito, 1991*; *Kubota and Taniguchi, 1998*), and we added a low-threshold T-type $Ca^{2+}$ current ($I_{CaT}$) conductance responsible for the post-inhibitory rebound firing seen in $HVC_X$ and $HVC_{INT}$ neurons (*Daou et al., 2013*). A small-conductance $Ca^{2+}$-activated $K^+$ current ($I_{SK}$) was added for $HVC_{RA}$ and $HVC_X$ neurons as it is responsible for the spike frequency adaptation feature that these two classes exhibit (*Daou et al., 2013*). For interneurons, we integrated a large magnitude of the delayed rectifier $K^+$ current conductance allowing these neurons to undershoot the resting membrane potential as seen experimentally (*Daou et al., 2013*; *Dutar et al., 1998*; *Kubota and Saito, 1991*; *Kubota and Taniguchi, 1998*). For $HVC_{RA}$ neurons, we added an A-type K+current that supports the delay to spiking seen in response to depolarizing current pulses (*Daou et al., 2013*; *Kubota and Taniguchi, 1998*; *Mooney and Prather, 2005*). High-threshold $Ca^{2+}$ conductance was added to all classes of HVC neurons (*Daou et al., 2013*; *Kubota and Saito, 1991*; *Long et al., 2010*). In total, the model was designed to include spike-producing currents ($I_K$ and $I_{Na}$), a high-threshold L-type $Ca^{2+}$ current ($I_{CaL}$), a low-threshold T-type $Ca^{2+}$ current ($I_{CaT}$), a small-conductance $Ca^{2+}$-activated $K^+$ current ($I_{SK}$), an A-type $K^+$ current ($I_A$), a hyperpolarization-activated current ($I_H$), and a leak current ($I_L$). The membrane potential of each HVC neuron obeys the following equations:

$$C_m \frac{dV_{RA}}{dt} = -I_L - I_K - I_{Na} - I_{CaL} - I_A - I_{SK} \tag{1}$$

$$C_m \frac{dV_X}{dt} = -I_L - I_K - I_{Na} - I_{CaL} - I_{CaT} - I_{SK} - I_H \tag{2}$$

$$C_m \frac{dV_{INT}}{dt} = -I_L - I_K - I_{Na} - I_{CaL} - I_{CaT} - I_H \tag{3}$$

where $C_m$ is the membrane capacitance. The associated equations and parameters for each of the activation/inactivation gating variables for each ionic current are given in *Daou et al., 2013* and shown below. In total, every single model $HVC_{RA}$, $HVC_X$, and $HVC_{INT}$ neuron had a total of 6, 8, and 7 ODEs, respectively, that govern their intrinsic dynamics. Every synaptic current that was integrated to any model neuron added a new ODE to the set of ODEs governing the membrane potential of the corresponding model neuron. See Appendix 1 for detailed equations.

## Synaptic currents

In addition to the ionic currents above that orchestrate the internal dynamics of each HVC neuron, we integrated synaptic currents in order to reproduce the biological features of the voltage traces observed in vivo. Excitatory (AMPA) and inhibitory (GABA$_A$) synaptic currents were used to connect neurons inside each architecture based on the pharmacological dual synaptic connections as described by *Mooney and Prather, 2005*. Each synaptic current represents the synaptic input(s) from the presynaptic cell(s) to the particular HVC model neuron and is modeled as $I_{syn} = \sum_X I_{X \to Y}$ where $I_{X \to Y} = g_{X \to Y} s_{X \to Y} (V - V_{X \to Y})$. Here, the summation is taken over the presynaptic HVC neurons where X represents a presynaptic cell, Y represents a postsynaptic cell, $V_{X \to Y}$ is the reversal potential for the synapse in the postsynaptic cell with $V_{X \to Y} = V_{AMPA}$ for excitatory input and $V_{X \to Y} = V_{GABA-A}$ for inhibitory input.

The model equations for the synaptic currents are detailed in APPENDIX 1, taken after (*Destexhe et al., 1994*; *Varela et al., 1997*). We limited our synaptic currents' choices in all networks to AMPA (excitatory) and GABA$_A$ (inhibitory) without integrating NMDA and GABA$_B$ for the following reasons: (1) both AMPA and GABA$_A$ currents are voltage-dependent with simple activation kinetics that do not depend on further parameters that are very hard to tune or calibrate (for e.g. NMDA current relies on $Mg^{2+}$ concentration and GABA$_B$ on G-proteins dynamics, both of which require additional ODEs and series of parameters that we do not know about in the HVC), (2) adding the additional synaptic currents does not have a significant contribution to the network dynamics we are building because the emphasis is on excitation and inhibition, and we could convey the mechanisms we envision that

orchestrate each network with these two currents solely. Therefore, model $HVC_{RA}$ neurons send their excitatory afferents to other $HVC_{RA}$ neurons as well as to $HVC_{INT}$ neurons via AMPA currents. Model $HVC_{INT}$ neurons send their inhibitory afferents to both $HVC_{RA}$ and $HVC_X$ neurons via $GABA_A$ currents. And lastly, model $HVC_X$ neurons excite $HVC_{INT}$ neurons via AMPA currents.

## Desired network activity

Our aim here is to generate the optimal and desired network activity that's generated by the three classes of HVC neurons during singing. Therefore, we focused on model parameters and their underlying mechanisms that play key roles in (1) reproducing the patterns without breaking the sequence of activity propagation and (2) generating biologically realistic traces for each class as shown using intracellular recordings in vivo in a way to maintain spike shapes, burst patterns, rebound firing/bursting, subthreshold oscillations, etc … In a nutshell, the behavior of the network was considered desired and 'good' if the model voltage traces for the total populations in each of the three classes of HVC neurons in the network matched the following: (1) the time-locked and characteristic bursting of $HVC_{RA}$ neurons (3–6 spikes for a~10ms duration), with spikes riding on a plateau, (2) $HVC_X$ neurons eliciting 1–4 bursts (4–9 spikes per burst) that are also time-locked and that are mostly rebound bursts from inhibition (*Lewicki, 1996*; *Mooney, 2000*), (3) $HVC_{INT}$ neurons exhibiting tonic activation with spiking and bursting throughout the song, and (4) spike frequency, spike amplitude, sags and/or rebound upon inhibition, resting membrane potential, and other known features of the intrinsic properties of the three classes of HVC neurons exhibited for each of the classes that exhibit them (*Daou et al., 2013*).

## Maximal and synaptic conductance variations

Automated adjustment of model parameters was performed to qualitatively reproduce desired membrane potential trajectories, as described next. Fixed parameter values for HVC neurons used in the simulations are given in *Appendix 1—table 1*. Parameters that vary between the different model neurons are shown in *Figure 13A*.

Some maximal conductances were fixed while others were allowed to vary. We fixed $g_{Na}$ and $g_K$ for each class of the HVC neurons to values that had been shown earlier to accurately fit the spike morphologies (upstrokes and downstrokes of action potentials, plateaus, etc…) in response to applied current given in vitro (*Daou et al., 2013*). For example, $HVC_{INT}$ neurons' spikes exhibit a relatively large undershoot of the resting membrane potential, while $HVC_X$ and $HVC_{RA}$ spikes ride on a plateau with characteristic properties (*Daou et al., 2013*). We also fixed the value of $g_{CaL}$ because we could achieve the same accuracy of fitting by varying $g_{SK}$, and so could not distinguish between the two.

The four key conductances in our model that played crucial roles in controlling not only the intrinsic properties of the HVC neurons they're expressed in but also in shaping overall network activity and sequence propagation are $g_{SK}$, $g_h$, $g_A$, and $g_{CaT}$. As a recap, $g_{SK}$ is expressed in $HVC_X$ and $HVC_{RA}$, $g_H$ and $g_{CaT}$ in $HVC_X$ and $HVC_{INT}$, and $g_A$ in $HVC_{RA}$ only (*Daou et al., 2013*). Random variations in these four parameters were performed to qualitatively reproduce membrane potential trajectories of the three classes of HVC neurons as seen firing when the bird is singing.

In a previous study, *Daou and Margoliash, 2020* showed that intracellular recordings from X-projecting neurons in adult zebra finch brain slices share similar spike waveform morphologies, with modeling indicating similar magnitudes of their principal ion currents. To that end, we fixed in our network the intrinsic properties of the population of $HVC_X$ neurons to the same values. Therefore, the automated variations in the conductances held for $HVC_X$ neurons were done at the population level by varying a corresponding ionic conductance value (say, $g_{SK}$) and setting its value to all of the $HVC_X$ population. We also checked the effects of removing this constraint in *Figure 12*, that is, allowing the intrinsic properties of $HVC_X$ neurons to vary like other parameters.

We first manually selected the four key conductances to default values that generate the desired behavior of the network as described in the previous section. Network robustness to varied maximal conductances as well as the legitimate ranges for each maximal conductance was determined by simulating the network many times, each time with a random variation in the maximal conductance about its default value. The network response was considered accurate if the network generated sequential bursting and all of the desired features described earlier. The range of variation of the

randomly-varying parameter was increased or decreased randomly up until the point that the network ceased to be accurate.

The variation in parameters was done at the population level (setting the maximal conductance for all neurons of the same class to a single value and randomly changing that value for all), as well as on the individual neuronal level (varying the maximal conductance for one neuron at a time of the same class while fixing the others to their default values). This is different for the $HVC_X$ parameters, where the corresponding conductances at the population and the neuronal levels are considered the same since we assumed the same intrinsic properties as described earlier. If any of the simulations generated networks where any of the $HVC_{RA}$ neurons elicit bursts exhibiting spikes outside the 3–6 number of spikes range or duration of any burst longer than 10ms, then we ignore that parameter value. Similarly, if any of the simulations generate networks where any of the $HVC_X$ neurons eliciting more than 4 bursts, individual bursts exhibiting spikes outside the 4–9 spikes/burst range, then the corresponding parameters are ignored. Moreover, we also ignore parameters that generate unrealistic intrinsic properties of individual HVC neurons; for instance, if the resting membrane potentials of individual HVC neurons were outside the reported ranges (*Daou et al., 2013*), if $HVC_X$ and $HVC_{INT}$ neurons failed to generate sags and rebounds (depolarization or bursting) in response to inhibition, or if spikes' amplitudes were not realistic ($HVC_X$ and $HVC_{RA}$ spikes not riding on plateaus or $HVC_{INT}$ spikes not undershooting the RMP). We realize this might be inducing tough constraints on the selection of the parameters limiting the space for which the conductances are allowed to vary, but we opted for this method in order to generate the optimal ranges in which intrinsic and synaptic conductances are able to reproduce the biophysically realistic firing patterns seen during singing. We therefore ended up with lists of ranges for each conductance that was varied and in each class of HVC neuron, such that the desired network activity is maintained. *Figure 13A* shows the ranges for each maximal conductance that had been allowed to vary for the three classes of HVC neurons, while maintaining robust network propagation and biologically realistic in vivo behavior.

Similar to what was done with the maximal ionic conductances, we conducted automated and random variations for all synaptic conductances in the model (no synaptic conductance was fixed). *Figure 13B* reports the ranges of the synaptic conductances that were able to maintain the robustness of network propagation and the general in vivo-like desired behavior of all neuronal classes. All synaptic conductances showed considerable ranges during which sequential activity is propagated and the overall desired network activity is maintained, with the exception of the GABA conductance from $HVC_{INT}$ to $HVC_{RA}$ ($g_{GABA_{INT \to RA}}$). Increasing $g_{GABA_{INT \to RA}}$ to larger magnitudes would induce an inhibition in the $HVC_{RA}$ pushing its voltage below its resting membrane potential (due to the dense bursting and firing in $HVC_{INT}$), and this is not realistic because it's been shown that during singing $HVC_{RA}$ neurons ride on a depolarizing plateau throughout the song (*Long et al., 2010*). Moreover, while the intrinsic properties of $HVC_X$ neurons were set to the same values, the synaptic parameters associated with each $HVC_X$ neuron (afferent and efferent) were allowed to vary from one neuron to another.

Moreover, to account for synaptic variability, we introduced a stochastic input current of the form $I_{noise}(t) = \sigma . \xi(t)$ where $\xi(t)$ is a Gaussian white noise with zero mean and unit variance, and $\sigma$ is the noise amplitude. This stochastic drive was introduced to every model neuron and it mimics the fluctuations in synaptic input arising from random presynaptic activity and background noise. For values of $\sigma$ within 1–5% of the mean synaptic conductance, the stochastic current has no effect on network propagation. For larger values of $\sigma$, the desired network activity was disrupted or halted.

## Acknowledgements

Daniel Margoliash engaged in valuable conversations as these results developed. ZBD, MC, and AD conceptualized, design the research and constructed the network; ZBD performed simulations and prepared figures, ZBD and AD interpreted results of simulations and drafted manuscript. This work was supported in part by grants provided to AD by the University Research Board and Farouk Jabr Foundation at the American University of Beirut.

## Additional information

### Funding

| Funder | Grant reference number | Author |
|---|---|---|
| Farouk Jabr Foundation | | Arij Daou |
| University Research Board | | Arij Daou |

The funders had no role in study design, data collection and interpretation, or the decision to submit the work for publication.

### Author contributions

Zeina Bou Diab, Data curation, Software, Formal analysis, Visualization, Methodology, Writing – original draft; Marc Chammas, Data curation, Formal analysis, Methodology; Arij Daou, Conceptualization, Supervision, Funding acquisition, Validation, Writing – original draft, Project administration, Writing - review and editing

### Author ORCIDs

Zeina Bou Diab ⓘ https://orcid.org/0009-0000-4195-006X
Arij Daou ⓘ https://orcid.org/0000-0001-7590-4774

Reviewer #2 (Public review): https://doi.org/10.7554/eLife.105526.3.sa1
Author response https://doi.org/10.7554/eLife.105526.3.sa2

## Additional files

### Supplementary files

MDAR checklist

Source code 1. This code connects all neurons in the network via the corresponding ionic currents.

Source code 2. This function builds the template of projecting neurons as well as interneurons to be simulated in the network.

Source code 3. This is the differential equations file containing the corresponding equations, this file is called in all simulations.

### Data availability

All figures generated in this study are included in the manuscript. Source codes have been provided for *Figures 2, 4–12*.

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

## Appendix 1

### Voltage-gated ionic currents

The constant-conductance leak current is $I_L = g_L (V - V_L)$. The remaining voltage-gated ionic currents have non-constant dependent currents with activation/inactivation kinetics:

$$I_K = g_K n^4 (V - V_K) \tag{4}$$

$$I_{Na} = g_{Na} m^3_\infty (V) h (V - V_{Na}) \tag{5}$$

$$I_A = g_A a_\infty (V) e (V - V_K) \tag{6}$$

$$I_{CaL} = g_{Ca} s^2_\infty (V) \frac{Ca_{ex}}{1 - e^{\frac{2FV}{RT}}} \tag{7}$$

$$\text{where } x_\infty(V) = \frac{1}{1 + e^{\frac{V - \theta_x}{\sigma_x}}}, \quad x = m, a \text{ or } s \tag{8}$$

$$\text{and } \frac{dx}{dt} = \frac{x_\infty (V) - V}{\tau_x}, \ x = n, h \text{ or } e \tag{9}$$

where $x_\infty (V)$ for n and e is given by (8) and for $h_\infty$ as follows

$$h_\infty (V) = \frac{\alpha_h (V)}{\alpha_h (V) + \beta_h (V)} \tag{10}$$

$$\text{where } \alpha_h (V) = 0.128 e^{\frac{V + 15}{-18}} \tag{11}$$

$$\text{and } \beta_h (V) = \frac{4}{1 + e^{\frac{V + 27}{-5}}} \tag{12}$$

### Low-voltage activated T-type calcium current

The low-voltage activated T-type Ca2+current is described by the Goldman-Hodgkin-Katz formula:

$$I_{CaT} = g_{CaT} (a_T)^3_\infty (V) (b_T)^2_\infty (r_T) \frac{Ca_{ex}}{1 - e^{\frac{2FV}{RT}}} \tag{13}$$

$$\text{where } a_{T_\infty} (V) = \frac{1}{1 + e^{\frac{V - \theta_{a_T}}{\sigma_{a_T}}}} \tag{14}$$

$$\text{and } b_{T_\infty} (r_T) = \frac{1}{1 + e^{\frac{r_T - \theta_b}{\sigma_b}}} - \frac{1}{1 + e^{\frac{-\theta_b}{\sigma_b}}} \tag{15}$$

$$\text{with } \frac{dr_T}{dt} = \frac{r_{T_\infty} (V) - r_T}{\tau_{r_T} (V)} \tag{16}$$

$$\text{and } r_{T_\infty}(V) = \frac{1}{1 + e^{\frac{V - \theta_{a_T}}{\sigma_{a_T}}}} \tag{17}$$

$$\text{and } \tau_{r_T}(V) = \tau_{r_0} + \frac{\tau_{r_1}}{1 + e^{\frac{V - \theta_{r_{r_T}}}{\sigma_{r_{r_T}}}}} \tag{18}$$

## Calcium-dependent potassium current

The small conductance potassium current ($I_{SK}$) is modeled as

$$I_{SK} = g_{SK} k_\infty \left([Ca^{2+}]_i\right) (V - V_K) \tag{19}$$

$$\text{where } k_\infty \left([Ca^{2+}]_i\right) = \frac{[Ca^{2+}]_i^2}{[Ca^{2+}]_i^2 + k_s^2} \tag{20}$$

$$\text{and } \frac{d[Ca^{2+}]_i}{dt} = -f\varepsilon \left(I_{CaL} + I_{CaT}\right) + k_{Ca} \left([Ca^{2+}]_i - b_{Ca}\right) \tag{21}$$

## Hyperpolarization-activated inward current

The hyperpolarization-activated inward current's activation is modeled as in **Destexhe and Babloyantz, 1993** using a fast component ($r_f$) and a slow component ($r_s$) as follows:

$$I_H = g_H \left[k_r r_f + (1 - k_r) r_s\right] (V - V_h) \tag{22}$$

The fast activation component is given by:

$$\frac{dr_f}{dt} = \frac{r_{f_\infty}(V) - r_f}{\tau_{r_f}(V)} \tag{23}$$

$$\text{where } r_{f_\infty}(V) = \frac{1}{1 + e^{\frac{V - \theta_{r_f}}{\sigma_{r_f}}}} \tag{24}$$

$$\text{with its time constant } \tau_{r_f}(V) = \frac{p_{r_f}}{\frac{-7.4(V+70)}{e^{\left(\frac{V+70}{-0.8}\right)} - 1} + 65 \, e^{\left(\frac{V+56}{-23}\right)}} \tag{25}$$

The slow activation component obeys:

$$\frac{dr_s}{ds} = \frac{r_{s_\infty}(V) - r_s}{\tau_{r_s}} \tag{26}$$

$$\text{where } r_{s_\infty}(V) = \frac{1}{1 + e^{\frac{-(V - \theta_{r_s})}{\sigma_{r_s}}}} \tag{27}$$

Synaptic currents equations

$$I_{AMPA} = \overline{g_{AMPA}} s_{AMPA} (V - V_{AMPA})$$

$$I_{GABA-A} = \overline{g_{GABA-A}} s_{GABAA} (V - V_{GABA-A})$$

where $s_{AMPA}$ and $s_{GABAA}$ are given by

$$[T]_{pre} = \frac{T_{max}}{1 + \exp\left(\frac{V_{pre} - V_T}{K_p}\right)} \quad \text{and} \quad \frac{ds}{dt} = a_r[T](1 - s) - a_d s$$

with $T_{max} = 1$, $K_p = 5$, $V_T = 2$. For GABA$_A$, $a_r = 5$ and $a_d = 0.18$, while for AMPA, $a_r = 1.1$ and $a_d = 0.19$.

**Appendix 1—table 1.** Fixed parameter values used in all simulations.

| Parameter | Value | Parameter | Value |
|---|---|---|---|
| $V_L$ | −70 mV | $\theta_{a_T}$ | −64 mV |
| $V_K$ | −90 mV | $\theta_b$ | 0.4 mV |
| $V_{Na}$ | 50 mV | $\theta_{r_T}$ | −67 mV |
| $V_{Ca}$ | −85 mV | $\theta_{r_{r_T}}$ | 68 mV |
| $V_H$ | −30 mV | $\sigma_m$ | −5 mV |
| $g_L$ | 2 nS | $\sigma_n$ | −5 mV |
| $g_{Ca}$ | 19 nS | $\sigma_s$ | −0.05 mV |
| $\tau_n$ | 10 ms | $\sigma_a$ | −10 mV |
| $\tau_{hp}$ | 1000 ms | $\sigma_e$ | 5 mV |
| $\tau_e$ | 20 ms | $\sigma_{r_f}$ | 5 mV |
| $\tau_h$ | 1 ms | $\sigma_{r_s}$ | 25 mV |
| $\tau_{r_s}$ | 1500 ms | $\sigma_{a_T}$ | 7.8 mV |
| $\tau_{r_0}$ | 200 ms | $\sigma_b$ | −0.1 mV |
| $\tau_{r_1}$ | 87.5 ms | $\sigma_{r_T}$ | 2 mV |
| $\theta_m$ | −35 mV | $\sigma_{r_{r_T}}$ | 2.2 mV |
| $\theta_n$ | −30 mV | $f$ | 0.1 |
| $\theta_s$ | −20 mV | $\varepsilon$ | $0.0015\, pA^{-1}\, micro\, M\ msec^{-1}$ |
| $\theta_a$ | −20 mV | $k_{Ca}$ | $0.3\, msec^{-1}$ |
| $\theta_e$ | −60 mV | $b_{Ca}$ | $0.1\, micro\, M$ |
| $\theta_{r_f}$ | −105 mV | $k_s$ | $0.5\, micro\, M$ |
| $\theta_{r_s}$ | −105 mV | $p_{r_f}$ | 100 |
| $g_{Na_{HVC_{RA}}}$ | 300 $nS$ | $g_{K_{HVC_{RA}}}$ | 150 $nS$ |
| $g_{Na_{HVC_X}}$ | 450 $nS$ | $g_{K_{HVC_X}}$ | 80 $nS$ |
| $g_{Na_{HVC_{INT}}}$ | 800 $nS$ | $g_{K_{HVC_{INT}}}$ | 1200 $nS$ |

