## [Editor Report · eLife Assessment]

This computational study examines how neurons in the songbird premotor nucleus HVC might generate the precise, sparse burst sequences that drive adult song. The findings would be **useful** for understanding how intrinsic conductances and HVC microcircuitry may produce neural sequences, but the work is **incomplete** because of arbitrary network assumptions, insufficient consideration of biological details such as how silent gaps in song sequences are represented, and failure to incorporate interactions with auditory and brainstem inputs. As a result, the study offers limited advance and only a modest conceptual advance over prior models.

---

## [Referee Report · Reviewer #2 (Public review)]

Summary:

In this paper, the authors use numerical simulations to try to understand better a major experimental discovery in songbird neuroscience from 2002 by Richard Hahnloser and collaborators. The 2002 paper found that a certain class of projection neurons in the premotor nucleus HVC of adult male zebra finch songbirds, the neurons that project to another premotor nucleus RA, fired sparsely (once per song motif) and precisely (to about 1 ms accuracy) during singing.

The experimental discovery is important to understand since it initially suggested that the sparsely firing RA-projecting neurons acted as a simple clock that was localized to HVC and that controlled all details of the temporal hierarchy of singing: notes, syllables, gaps, and motifs. Later experiments suggested that the initial interpretation might be incomplete: that the temporal structure of adult male zebra finch songs instead emerged in a more complicated and distributed way, still not well understood, from the interaction of HVC with multiple other nuclei, including auditory and brainstem areas. So at least two major questions remain unanswered more than two decades after the 2002 experiment: What is the neurobiological mechanism that produces the sparse precise bursting: is it a local circuit in HVC or is it some combination of external input to HVC and local circuitry? And how is the sparse precise bursting in HVC related to a songbird's vocalizations?

The authors only investigate part of the first question, whether the mechanism for sparse precise bursts is local to HVC. They do so indirectly, by using conductance-based Hodgkin-Huxley-like equations to simulate the spiking dynamics of a simplified network that includes three known major classes of HVC neurons and such that all neurons within a class are assumed to be identical. A strength of the calculations is that the authors include known biophysically deduced details of the different conductances of the three majors classes of HVC neurons, and they take into account what is known, based on sparse paired recordings in slices, about how the three classes connect to one another. One weakness of the paper is that the authors make arbitrary and not-well-motivated assumptions about the network geometry, and they do not use the flexibility of their simulations to study how their results depend on their network assumptions. A second weakness is that they ignore many known experimental details such as projections into HVC from other nuclei, dendritic computations (the somas and dendrites are treated by the authors as point-like isopotential objects), the role of neuromodulators, and known heterogeneity of the interneurons. These weaknesses make it difficult for readers to know the relevance of the simulations for experiments and for advancing theoretical understanding.

Strengths:

The authors use conductance-based Hodgkin-Huxley-like equations to simulate spiking activity in a network of neurons intended to model more accurately songbird nucleus HVC of adult male zebra finches. Spiking models are much closer to experiments than models based on firing rates or on 2-state neurons.

The authors include information deduced from modeling experimental current-clamp data such as the types and properties of conductances. They also take into account how neurons in one class connect to neurons in other classes via excitatory or inhibitory synapses, based on sparse paired recordings in slices by other researchers.

The authors obtain some new results of modest interest such as how changes in the maximum conductances of four key channels (e.g., A-type K+ currents or Ca-dependent K+ currents) influence the structure and propagation of bursts, while simultaneously being able to mimic accurately current-clamp voltage measurements.

Weaknesses:

One weakness of this paper is the lack of a clearly stated, interesting, and relevant scientific question to try to answer. The authors do not discuss adequately in their introduction what questions have recent experimental and theoretical work failed to explain adequately concerning HVC neural dynamics and its role in producing vocalizations. The authors do not discuss adequately why they chose the approach of their paper and how their results address some of these questions.

For example, the authors need to explain in more detail how their calculations relate to the works of Daou et al, J. Neurophys. 2013 (which already fitted spiking models to neuronal data and identified certain conductances), to Jin et al J. Comput. Neurosci. 2007 (which already discussed how to get bursts using some experimental details), and to the rather similar paper by E. Armstrong and H. Abarbanel, J. Neurophys 2016, which already postulated and studied sequences of microcircuits in HVC. This last paper is not even cited by the authors.

The authors' main achievement is to show that simulations of a certain simplified and idealized network of spiking neurons, that includes some experimental details but ignores many others, can match some experimental results like current-clamp-derived voltage time series for the three classes of HVC neurons (although this was already reported in earlier work by Daou and collaborators in 2013), and simultaneously the robust propagation of bursts with properties similar to those observed in experiments. The authors also present results about how certain neuronal details and burst propagation change when certain key maximum conductances are varied.

But these are weak conclusions for two reasons. First, the authors did not do enough calculations to allow the reader to understand how many parameters were needed to obtain these fits and whether simpler circuits, say with fewer parameters and simpler network topology, could do just as well. Second, many previous researchers have demonstrated robust burst propagation in a variety of feed-forward models. So what is new and important about the authors' results compared to the previous computational papers?

Also missing is a discussion, or at least an acknowledgement, of the fact that not all of the fine experimental details of undershoots, latencies, spike structure, spike accommodation, etc may be relevant for understanding vocalization. While it is nice to know that some model can match these experimental details and produce realistic bursts, that does not mean that all of these details are relevant for the function of producing precise vocalizations. Scientific insights in biology often require exploring which of the many observed details can be ignored, and especially identifying the few that are essential for answering some questions. As one example, if HVC-X neurons are completely removed from the authors' model, does one still get robust and reasonable burst propagation of HVC-RA neurons? While part of nucleus HVC acts as a premotor circuit that drives nucleus RA, part of HVC is also related to learning. It is not clear that HVC-X neurons, which carry out some unknown calculation and transmit information to area X in a learning pathway, are relevant for burst production and propagation of HVC-RA neurons, and so relevant for vocalization. Simulations provide a convenient and direct way to explore questions of this kind.

One key question to answer is whether the bursting of HVC-RA projection neurons is based on a mechanism local to HVC or is some combination of external driving (say from auditory nuclei) and local circuitry. The authors do not contribute to answering this question because they ignore external driving and assume that the mechanism is some kind of intrinsic feed-forward circuit, which they put in by hand in a rather arbitrary and poorly justified way, by assuming the existence of small microcircuits consisting of a few HVC-RA, HVC-X, and HVC-I neurons that somehow correspond to "sub-syllabic segments". To my knowledge, experiments do not suggest the existence of such microcircuits nor does theory suggest the need for such microcircuits.

Another weakness of this paper is an unsatisfactory discussion of how the model was obtained, validated, and simulated. The authors should state as clearly as possible, in one location such as an appendix, what is the total number of independent parameters for the entire network and how parameter values were deduced from data or assigned by hand. With enough parameters and variables, many details can be fit arbitrarily accurately so researchers have to be careful to avoid overfitting. If parameter values were obtained by fitting to data, the authors should state clearly what was the fitting algorithm (some iterative nonlinear method, whose results can depend on the initial choice of parameters), what was the error function used for fitting (sum of least squares?), and what data were used for the fitting.

The authors should also state clearly what is the dynamical state of the network, the vector of quantities that evolve over time. (What is the dimension of that vector, which is also the number of ordinary differential equations that have to be integrated?) The authors do not mention what initial state was used to start the numerical integrations, whether transient dynamics were observed and what were their properties, or how the results depend on the choice of initial state. The authors do not discuss how they determined that their model was programmed correctly (it is difficult to avoid typing errors when writing several pages or more of a code in any language) or how they determined the accuracy of the numerical integration method beyond fitting to experimental data, say by varying the time step size over some range or by comparing two different integration algorithms.

Also disappointing is that the authors do not make any predictions to test, except rather weak ones such as that varying a maximum conductance sufficiently (which might be possible by using dynamic clamps) might cause burst propagation to stop or change its properties. Based on their results, the authors do not make suggestions for further experiments or calculations, but they should.

Comments on revised version:

The second version, unfortunately, did not address most of the substantive comments so that, while some parts of the discussion were expanded, most of the serious scientific weaknesses mentioned in the first round of review remain. The revised preprint is not a substantive improvement over the first.

---

## [Author Response]

The following is the authors’ response to the original reviews.

**Reviewer #1 (Public review):**
Summary:The paper presents a model for sequence generation in the zebra finch HVC, which adheres to cellular properties measured experimentally. However, the model is fine-tuned and exhibits limited robustness to noise inherent in the inhibitory interneurons within the HVC, as well as to fluctuations in connectivity between neurons. Although the proposed microcircuits are introduced as units for sub-syllabic segments (SSS), the backbone of the network remains a feedforward chain of HVC_RA neurons, similar to previous models.Strengths:The model incorporates all three of the major types of HVC neurons. The ion channels used and their kinetics are based on experimental measurements. The connection patterns of the neurons are also constrained by the experiments.Weaknesses:The model is described as consisting of micro-circuits corresponding to SSS. This presentation gives the impression that the model's structure is distinct from previous models, which connected HVC_RA neurons in feedforward chain networks (Jin et al 2007, Li & Greenside, 2006; Long et al 2010; Egger et al 2020). However, the authors implement single HVC_RA neurons into chain networks within each micro-circuit and then connect the end of the chain to the start of the chain in the subsequent micro-circuit. Thus, the HVC_RA neuron in their model forms a single-neuron chain. This structure is essentially a simplified version of earlier models.In the model of the paper, the chain network drives the HVC_I and HVC_X neurons. The role of the micro-circuits is more significant in organizing the connections: specifically, from HVC_RA neurons to HVC_I neurons, and from HVC_I neurons to both HVC_X and HVC_RA neurons.

We thank Reviewer 1 for their thoughtful comments.

While the reviewer is correct about the fact that the propagation of sequential activity in this model is primarily carried by HVC_RA_ neurons in a feed-forward manner, we need to emphasize that this is true only if there is no intrinsic or synaptic perturbation to the HVC network. For example, we showed in Figures 10 and 12 how altering the intrinsic properties of HVC_X_ neurons or for interneurons disrupts sequence propagation. In other words, while HVC_RA_ neurons are the key forces to carry the chain forward, the interplay between excitation and inhibition in our network as well as the intrinsic parameters for all classes of HVC neurons are equally important forces in carrying the chain of activity forward. Thus, the stability of activity propagation necessary for song production depend on a finely balanced network of HVC neurons, with all classes contributing to the overall dynamics. Moreover, all existing models that describe premotor sequence generation in the HVC either assume a distributed model (Elmaleh et al., 2021) that dictates that local HVC circuitry is not sufficient to advance the sequence but rather depends upon moment to-moment feedback through Uva (Hamaguchi et al., 2016), or assume models that rely on intrinsic connections within HVC to propagate sequential activity. In the latter case, some models assume that HVC is composed of multiple discrete subnetworks that encode individual song elements (Glaze & Troyer, 2013; Long & Fee, 2008; Wang et al., 2008), but lacks the local connectivity to link the subnetworks, while other models assume that HVC may have sufficient information in its intrinsic connections to form a single continuous network sequence (Long et al. 2010). The HVC model we present extends the concept of a feedforward network by incorporating additional neuronal classes that influence the propagation of activity (interneurons and HVC_X_ neurons). We have shown that any disturbance of the intrinsic or synaptic conductances of these latter neurons will disrupt activity in the circuit even when HVC_RA_ neurons properties are maintained.

In regard to the similarities between our model and earlier models, several aspects of our model distinguish it from prior work. In short, while several models of how sequence is generated within HVC have been proposed (Cannon et al., 2015; Drew & Abbott, 2003; Egger et al., 2020; Elmaleh et al., 2021; Galvis et al., 2018; Gibb et al., 2009a, 2009b; Hamaguchi et al., 2016; Jin, 2009; Long & Fee, 2008; Markowitz et al., 2015), all the models proposed either rely on intrinsic HVC circuitry to propagate sequential activity, rely on extrinsic feedback to advance the sequence or rely on both. These models do not capture the complex details of spike morphology, do not include the right ionic currents, do not incorporate all classes of HVC neurons, or do not generate realistic firing patterns as seen in vivo. Our model is the first biophysically realistic model that incorporates all classes of HVC neurons and their intrinsic properties. We tuned the intrinsic and the synaptic properties bases on the traces collected by Daou et al. (2013) and Mooney and Prather (2005) as shown in Figure 3. The three classes of model neurons incorporated to our network as well as the synaptic currents that connect them are based on Hodgkin- Huxley formalisms that contain ion channels and synaptic currents which had been pharmacologically identified. This is an advancement over prior models that primarily focused on the role of synaptic interactions or external inputs. The model is based on feedforward chain of microcircuits that encode for the different sub-syllabic segments and that interact with each other through structured feedback inhibition, defining an ordered sequence of cell firing. Moreover, while several models highlight the critical role of inhibitory interneurons in shaping the timing and propagation of bursts of activity in HVC_RA_ neurons, our work offers an intricate and comprehensive model that help understand this critical role played by inhibition in shaping song dynamics and ensuring sequence propagation.

How useful is this concept of micro-circuits? HVC neurons fire continuously even during the silent gaps. There are no SSS during these silent gaps.

Regarding the concern about the usefulness of the 'microcircuit' concept in our study, we appreciate the comment and we are glad to clarify its relevance in our network. While we acknowledge that HVC_RA_ neurons interconnect microcircuits, our model's dynamics are still best described within the framework of microcircuitry particularly due to the firing behavior of HVC_X_ neurons and interneurons. Here, we are referring to microcircuits in a more functional sense, rather than rigid, isolated spatial divisions (Cannon et al. 2015), and we now make this clear on page 21. A microcircuit in our model reflects the local rules that govern the interaction between all HVC neuron classes within the broader network, and that are essential for proper activity propagation. For example, HVC_INT_ neurons belonging to any microcircuit burst densely and at times other than the moments when the corresponding encoded SSS is being “sung”. What makes a particular interneuron belong to this microcircuit or the other is merely the fact that it cannot inhibit HVC_RA_ neurons that are housed in the microcircuit it belongs to. In particular, if HVC_INT_ inhibits HVC_RA_ in the same microcircuit, some of the HVC_RA_ bursts in the microcircuit might be silenced by the dense and strong HVC_INT_ inhibition breaking the chain of activity again. Similarly, HVC_X_ neurons were selected to be housed within microcircuits due to the following reason: if an HVC_X_ neuron belonging to microcircuit i sends excitatory input to an HVC_INT_ neuron in microcircuit j, and that interneuron happens to select an HVC_RA_ neuron from microcircuit i, then the propagation of sequential activity will halt, and we’ll be in a scenario similar to what was described earlier for HVC_INT_ neurons inhibiting HVC_RA_ neurons in the same microcircuit.

We agree that there are no sub-syllabic segments described during the silent gaps and we thank the reviewer to pointing this out. Although silent gaps are integral to the overall process of song production, we have not elaborated on them in this model due to the lack of a clear, biophysically grounded representation for the gaps themselves at the level of HVC. Our primary focus has been on modeling the active, syllable-producing phases of the song, where the HVC network’s sequential dynamics are critical for song. However, one can think the encoding of silent gaps via similar mechanisms that encode SSSs, where each gap is encoded by similar microcircuits comprised of the three classes of HVC neurons (let’s call them GAP rather than SSS) that are active only during the silent gaps. In this case, the propagation of sequential activity is carried throughout the GAPs from the last SSS of the previous syllable to the first SSS of the subsequent syllable. This is no described more clearly on page 22 of the manuscript.

A significant issue of the current model is that the HVC_RA to HVC_RA connections require fine-tuning, with the network functioning only within a narrow range of g_AMPA (Figure 2B). Similarly, the connections from HVC_I neurons to HVC_RA neurons also require fine-tuning. This sensitivity arises because the somatic properties of HVC_RA neurons are insufficient to produce the stereotypical bursts of spikes observed in recordings from singing birds, as demonstrated in previous studies (Jin et al 2007; Long et al 2010). In these previous works, to address this limitation, a dendritic spike mechanism was introduced to generate an intrinsic bursting capability, which is absent in the somatic compartment of HVC_RA neurons. This dendritic mechanism significantly enhances the robustness of the chain network, eliminating the need to fine-tune any synaptic conductances, including those from HVC_I neurons (Long et al 2010). Why is it important that the model should NOT be sensitive to the connection strengths?

We thank the reviewer for the comment. While mathematical models designed for highly complex nonlinear biological processes tangentially touch the biological realism, the current network as is right now is the first biologically realistic-enough network model designed for HVC that explains sequence propagation. We do not include dendritic processes in our network although that increases the realistic dynamics for various reasons. (1) The ion channels we integrated into the somatic compartment are known pharmacologically (Daou et al. 2013), but we don’t know about the dendritic compartment’s intrinsic properties of HVC neurons and the cocktail of ion channels that are expressed there. (2) We are able to generate realistic bursting in HVC_RA_ neurons despite the single compartment, and the main emphasis in this network is on the interactions between excitation and inhibition, the effects of ion channels in modulating sequence propagation, etc … (3) The network model already incorporates thousands of ODEs that govern the dynamics of each of the HVC neurons, so we did not want to add more complexity to the network especially that we don’t know the biophysical properties of the dendritic compartments.

Therefore, our present focus is on somatic dynamics and the interaction between HVC_RA_ and HVC_INT_ neurons, but we acknowledge the importance of these processes in enhancing network resiliency. Although we agree that adding dendritic processes improves robustness, we still think that somatic processes alone can offer insightful information on the sequential dynamics of the HVC network. While the network should be robust across a wide range of parameters, it is also essential that certain parameters are designed to filter out weaker signals, ensuring that only reliable, precise patterns of activity propagate. Hence, we specifically chose to make the HVC_RA_-to-HVC_RA_ excitatory connections more sensitive (narrow range of values) such that only strong, precise and meaningful stimuli can propagate through the network representing the high stereotypy and precision seen in song production.

First, the firing of HVC_I neurons is highly noisy and unreliable. HVC_I neurons fire spontaneous, random spikes under baseline conditions. During singing, their spike timing is imprecise and can vary significantly from trial to trial, with spikes appearing or disappearing across different trials. As a result, their inputs to HVC_RA neurons are inherently noisy. If the model relies on precisely tuned inputs from HVC_I neurons, the natural fluctuations in HVC_I firing would render the model non-functional. The authors should incorporate noisy HVC_I neurons into their model to evaluate whether this noise would render the model non-functional.

We acknowledge that under baseline and singing settings, interneurons fire in an extremely noisy and inaccurate manner, although they exhibit time locked episodes in their activity (Hahnloser et al 2002, Kozhinikov and Fee 2007). In order to mimic the biological variability of these neurons, our model does, in fact, include a stochastic current to reflect the intrinsic noise and random variations in interneuron firing shown in vivo (and we highlight this in the Methods). However, to make sure the network is resilient to this randomness in interneuron firing, introduced a stochastic input current of the form I_noise_ (t) = σ.ξ(t) where ξ(t) is a Gaussian white noise with zero mean and unit variance, and σ is the noise amplitude. This stochastic drive was introduced to every model neuron and it mimics the fluctuations in synaptic input arising from random presynaptic activity and background noise. For values of σ within 1-5% of the mean synaptic conductance, the stochastic current has no effect on network propagation. For larger values of σ, the desired network activity was disrupted or halted. We now talk about this on page 22 of the manuscript.

Second, Kosche et al. (2015) demonstrated that reducing inhibition by suppressing HVC_I neuron activity makes HVC_RA firing less sparse but does not compromise the temporal precision of the bursts. In this experiment, the local application of gabazine should have severely disrupted HVC_I activity. However, it did not affect the timing precision of HVC_RA neuron firing, emphasizing the robustness of the HVC timing circuit. This robustness is inconsistent with the predictions of the current model, which depends on finely tuned inputs and should, therefore, be vulnerable to such disruptions.

We thank the reviewer for the comment. The differences between the Kosche et al. (2015) findings and the predictions of our model arise from differences in the aspect of HVC function we are modeling. Our model is more sensitive to inhibition, which is a designed mechanism for achieving precise song patterning. This is a modeling simplification we adopted to capture specific characteristics of HVC function. Hence, Kosche et al. (2015) findings do not invalidate the approach of our model, but highlights that HVC likely operates with several**,** redundant mechanisms that overall ensure temporal precision**.**

Third, the reliance on fine-tuning of HVC_RA connections becomes problematic if the model is scaled up to include groups of HVC_RA neurons forming a chain network, rather than the single HVC_RA neurons used in the current work. With groups of HVC_RA neurons, the summation of presynaptic inputs to each HVC_RA neuron would need to be precisely maintained for the model to function. However, experimental evidence shows that the HVC circuit remains functional despite perturbations, such as a few degrees of cooling, micro-lesions, or turnover of HVC_RA neurons. Such robustness cannot be accounted for by a model that depends on finely tuned connections, as seen in the current implementation.

Our model of individual HVC_RA_ neurons and as stated previously is reductive model that focuses on understanding the mechanisms that govern sequential neural activity. We agree that scaling the model to include many of HVC_RA_ neurons poses challenges, specifically concerning the summation of presynaptic inputs. However, our model can still be adapted to a larger network without requiring the level of fine-tuning currently needed. In fact, the current fine-tuning of synaptic connections in the model is a reflection of fundamental network mechanisms rather than a limitation when scaling to a larger network. Besides, one important feature of this neural network is redundancy. Even if some neurons or synaptic connections are impaired, other neurons or pathways can compensate for these changes, allowing the activity propagation to remain intact.

The authors examined how altering the channel properties of neurons affects the activity in their model. While this approach is valid, many of the observed effects may stem from the delicate balancing required in their model for proper function. In the current model, HVC_X neurons burst as a result of rebound activity driven by the I_H current. Rebound bursts mediated by the I_H current typically require a highly hyperpolarized membrane potential. However, this mechanism would fail if the reversal potential of inhibition is higher than the required level of hyperpolarization. Furthermore, Mooney (2000) demonstrated that depolarizing the membrane potential of HVC_X neurons did not prevent bursts of these neurons during forward playback of the bird's own song, suggesting that these bursts (at least under anesthesia, which may be a different state altogether) are not necessarily caused by rebound activity. This discrepancy should be addressed or considered in the model.

In our HVC network model, one goal with HVC_X_ neurons is to generate bursts in their underlying neuron population. Since HVC_X_ neurons in our model receive only inhibitory inputs from interneurons, we rely on inhibition followed by rebound bursts orchestrated by the I_H_ and the I_CaT_ currents to achieve this goal. The interplay between the T-type Ca^++^ current and the H current in our model is fundamental to generate their corresponding bursts, as they are sufficient for producing the desired behavior in the network. Due to this interplay, we do not need significant inhibition to generate rebound bursts, because the T-type Ca_++_ current’s conductance can be stronger leading to robust rebound bursting even when the degree of inhibition is not very strong. This is now highlighted on page 42 in the revised version.

Some figures contain direct copies of figures from published papers. It is perhaps a better practice to replace them with schematics if possible.

We wanted on purpose to keep the results shown in Mooney and Prather (2005) to be shown as is, in order to compare them with our model simulations highlighting the degree of resemblance. We believe that creating schematics of the Mooney and Prather (2005) results will not have the same impact, similarly creating a schematic for Hahnloser et al (2002) results won’t help much. However, if the reviewer still believes that we should do that, we’re happy to do it.

**Reviewer #2 (Public review):**
Summary:In this paper, the authors use numerical simulations to try to understand better a major experimental discovery in songbird neuroscience from 2002 by Richard Hahnloser and collaborators. The 2002 paper found that a certain class of projection neurons in the premotor nucleus HVC of adult male zebra finch songbirds, the neurons that project to another premotor nucleus RA, fired sparsely (once per song motif) and precisely (to about 1 ms accuracy) during singing.The experimental discovery is important to understand since it initially suggested that the sparsely firing RA-projecting neurons acted as a simple clock that was localized to HVC and that controlled all details of the temporal hierarchy of singing: notes, syllables, gaps, and motifs. Later experiments suggested that the initial interpretation might be incomplete: that the temporal structure of adult male zebra finch songs instead emerged in a more complicated and distributed way, still not well understood, from the interaction of HVC with multiple other nuclei, including auditory and brainstem areas. So at least two major questions remain unanswered more than two decades after the 2002 experiment: What is the neurobiological mechanism that produces the sparse precise bursting: is it a local circuit in HVC or is it some combination of external input to HVC and local circuitry? And how is the sparse precise bursting in HVC related to a songbird's vocalizations? The authors only investigate part of the first question, whether the mechanism for sparse precise bursts is local to HVC. They do so indirectly, by using conductance-based Hodgkin-Huxley-like equations to simulate the spiking dynamics of a simplified network that includes three known major classes of HVC neurons and such that all neurons within a class are assumed to be identical. A strength of the calculations is that the authors include known biophysically deduced details of the different conductances of the three major classes of HVC neurons, and they take into account what is known, based on sparse paired recordings in slices, about how the three classes connect to one another. One weakness of the paper is that the authors make arbitrary and not well-motivated assumptions about the network geometry, and they do not use the flexibility of their simulations to study how their results depend on their network assumptions. A second weakness is that they ignore many known experimental details such as projections into HVC from other nuclei, dendritic computations (the somas and dendrites are treated by the authors as point-like isopotential objects), the role of neuromodulators, and known heterogeneity of the interneurons. These weaknesses make it difficult for readers to know the relevance of the simulations for experiments and for advancing theoretical understanding.Strengths:The authors use conductance-based Hodgkin-Huxley-like equations to simulate spiking activity in a network of neurons intended to model more accurately songbird nucleus HVC of adult male zebra finches. Spiking models are much closer to experiments than models based on firing rates or on 2-state neurons.The authors include information deduced from modeling experimental current-clamp data such as the types and properties of conductances. They also take into account how neurons in one class connect to neurons in other classes via excitatory or inhibitory synapses, based on sparse paired recordings in slices by other researchers. The authors obtain some new results of modest interest such as how changes in the maximum conductances of four key channels (e.g., A-type K+ currents or Ca-dependent K+ currents) influence the structure and propagation of bursts, while simultaneously being able to mimic accurately current-clamp voltage measurements.Weaknesses:One weakness of this paper is the lack of a clearly stated, interesting, and relevant scientific question to try to answer. In the introduction, the authors do not discuss adequately which questions recent experimental and theoretical work have failed to explain adequately, concerning HVC neural dynamics and its role in producing vocalizations. The authors do not discuss adequately why they chose the approach of their paper and how their results address some of these questions.For example, the authors need to explain in more detail how their calculations relate to the works of Daou et al, J. Neurophys. 2013 (which already fitted spiking models to neuronal data and identified certain conductances), to Jin et al J. Comput. Neurosci. 2007 (which already discussed how to get bursts using some experimental details), and to the rather similar paper by E. Armstrong and H. Abarbanel, J. Neurophys 2016, which already postulated and studied sequences of microcircuits in HVC. This last paper is not even cited by the authors.

We thank the reviewer for this valuable comment, and we agree that we did not clarify enough throughout the paper the utility of our model or how it advanced our understanding of the HVC dynamics and circuitry. To that end, we revised several places of the manuscript and made sure to cite and highlight the relevance and relatedness of the mentioned papers.

In short, and as mentioned to Reviewer 1, while several models of how sequence is generated within HVC have been proposed (Cannon et al., 2015; Drew & Abbott, 2003; Egger et al., 2020; Elmaleh et al., 2021; Galvis et al., 2018; Gibb et al., 2009a, 2009b; Hamaguchi et al., 2016; Jin, 2009; Long & Fee, 2008; Markowitz et al., 2015; Jin et al., 2007), all the models proposed either rely on intrinsic HVC circuitry to propagate sequential activity, rely on extrinsic feedback to advance the sequence or rely on both. These models do not capture the complex details of spike morphology, do not include the right ionic currents, do not incorporate all classes of HVC neurons, or do not generate realistic firing patterns as seen in vivo. Our model is the first biophysically realistic model that incorporates all classes of HVC neurons and their intrinsic properties.

No existing hypothesis had been challenged with our model, rather; our model is a distillation of the various models that’s been proposed for the HVC network. We go over this in detail in the Discussion. We believe that the network model we developed provide a step forward in describing the biophysics of HVC circuitry, and may throw a new light on certain dynamics in the mammalian brain, particularly the motor cortex and the hippocampus regions where precisely-timed sequential activity is crucial. We suggest that temporally-precise sequential activity may be a manifestation of neural networks comprised of chain of microcircuits, each containing pools of excitatory and inhibitory neurons, with local interplay among neurons of the same microcircuit and global interplays across the various microcircuits, and with structured inhibition as well as intrinsic properties synchronizing the neuronal pools and stabilizing timing within a firing sequence.

The authors' main achievement is to show that simulations of a certain simplified and idealized network of spiking neurons, which includes some experimental details but ignores many others, match some experimental results like current-clamp-derived voltage time series for the three classes of HVC neurons (although this was already reported in earlier work by Daou and collaborators in 2013), and simultaneously the robust propagation of bursts with properties similar to those observed in experiments. The authors also present results about how certain neuronal details and burst propagation change when certain key maximum conductances are varied. However, these are weak conclusions for two reasons. First, the authors did not do enough calculations to allow the reader to understand how many parameters were needed to obtain these fits and whether simpler circuits, say with fewer parameters and simpler network topology, could do just as well. Second, many previous researchers have demonstrated robust burst propagation in a variety of feed-forward models. So what is new and important about the authors' results compared to the previous computational papers?

A major novelty of our work is the incorporation of experimental data with detailed network models. While earlier works have established robust burst propagation, our model uses realistic ion channel kinetics and feedback inhibition not only to reproduce experimental neural activity patterns but also to suggest prospective mechanisms for song sequence production in the most biophysical way possible. This aspect that distinguishes our work from other feed-forward models. We go over this in detail in the Discussion. However, the reviewer is right regarding the details of the calculations conducted for the fits, we will make sure to highlight this in the Methods and throughout the manuscript with more details.

We believe that the network model we developed provide a step forward in describing the biophysics of HVC circuitry, and may throw a new light on certain dynamics in the mammalian brain, particularly the motor cortex and the hippocampus regions where precisely-timed sequential activity is crucial. We suggest that temporally-precise sequential activity may be a manifestation of neural networks comprised of chain of microcircuits, each containing pools of excitatory and inhibitory neurons, with local interplay among neurons of the same microcircuit and global interplays across the various microcircuits, and with structured inhibition as well as intrinsic properties synchronizing the neuronal pools and stabilizing timing within a firing sequence.

Also missing is a discussion, or at least an acknowledgment, of the fact that not all of the fine experimental details of undershoots, latencies, spike structure, spike accommodation, etc may be relevant for understanding vocalization. While it is nice to know that some models can match these experimental details and produce realistic bursts, that does not mean that all of these details are relevant for the function of producing precise vocalizations. Scientific insights in biology often require exploring which of the many observed details can be ignored and especially identifying the few that are essential for answering some questions. As one example, if HVC-X neurons are completely removed from the authors' model, does one still get robust and reasonable burst propagation of HVC-RA neurons? While part of the nucleus HVC acts as a premotor circuit that drives the nucleus RA, part of HVC is also related to learning. It is not clear that HVC-X neurons, which carry out some unknown calculation and transmit information to area X in a learning pathway, are relevant for burst production and propagation of HVCRA neurons, and so relevant for vocalization. Simulations provide a convenient and direct way to explore questions of this kind.One key question to answer is whether the bursting of HVC-RA projection neurons is based on a mechanism local to HVC or is some combination of external driving (say from auditory nuclei) and local circuitry. The authors do not contribute to answering this question because they ignore external driving and assume that the mechanism is some kind of intrinsic feed-forward circuit, which they put in by hand in a rather arbitrary and poorly justified way, by assuming the existence of small microcircuits consisting of a few HVC-RA, HVC-X, and HVC-I neurons that somehow correspond to "sub-syllabic segments". To my knowledge, experiments do not suggest the existence of such microcircuits nor does theory suggest the need for such microcircuits.

Recent results showed a tight correlation between the intrinsic properties of neurons and features of song (Daou and Margoliash 2020, Medina and Margoliash 2024), where adult birds that exhibit similar songs tend to have similar intrinsic properties. While this is relevant, we acknowledge that not all details may be necessary for every aspect of vocalization, and future models could simplify concentrate on core dynamics and exclude certain features while still providing insights into the primary mechanisms.

The question of whether HVC_X_ neurons are relevant for burst propagation given that our model includes these neurons as part of the network for completeness, the reviewer is correct, the propagation of sequential activity in this model is primarily carried by HVC_RA_ neurons in a feed-forward manner, but only if there is no perturbation to the HVC network. For example, we have shown how altering the intrinsic properties of HVC_X_ neurons or for interneurons disrupts sequence propagation. In other words, while HVC neurons are the key forces to carry the chain forward, the interplay between excitation and inhibition in our network as well as the intrinsic parameters for all classes of HVC neurons are equally important forces in carrying the chain of activity forward. Thus, the stability of activity propagation necessary for song production depend on a finely balanced network of HVC neurons, with all classes contributing to the overall dynamics.

We agree with the reviewer however that a potential drawback of our model is that its sole focus is on local excitatory connectivity within the HVC (Kornfeld et al., 2017; Long et al., 2010), while HVC neurons receive afferent excitatory connections (Akutagawa & Konishi, 2010; Nottebohm et al., 1982) that plays significant roles in their local dynamics. For example, the excitatory inputs that HVC neurons receive from Uvaeformis may be crucial in initiating (Andalman et al., 2011; Danish et al., 2017; Galvis et al., 2018) or sustaining (Hamaguchi et al., 2016) the sequential activity. While we acknowledge this limitation, our main contribution in this work is the biophysical insights onto how the patterning activity in HVC is largely shaped by the intrinsic properties of the individual neurons as well as the synaptic properties where excitation and inhibition play a major role in enabling neurons to generate their characteristic bursts during singing. This is true and holds irrespective of whether an external drive is injected onto the microcircuits or not. We elaborated on this further in the revised version in the Discussion.

Another weakness of this paper is an unsatisfactory discussion of how the model was obtained, validated, and simulated. The authors should state as clearly as possible, in one location such as an appendix, what is the total number of independent parameters for the entire network and how parameter values were deduced from data or assigned by hand. With enough parameters and variables, many details can be fit arbitrarily accurately so researchers have to be careful to avoid overfitting. If parameter values were obtained by fitting to data, the authors should state clearly what the fitting algorithm was (some iterative nonlinear method, whose results can depend on the initial choice of parameters), what the error function used for fitting (sum of least squares?) was, and what data were used for the fitting.The authors should also state clearly the dynamical state of the network, the vector of quantities that evolve over time. (What is the dimension of that vector, which is also the number of ordinary differential equations that have to be integrated?) The authors do not mention what initial state was used to start the numerical integrations, whether transient dynamics were observed and what were their properties, or how the results depended on the choice of the initial state. The authors do not discuss how they determined that their model was programmed correctly (it is difficult to avoid typing errors when writing several pages or more of a code in any language) or how they determined the accuracy of the numerical integration method beyond fitting to experimental data, say by varying the time step size over some range or by comparing two different integration algorithms.

We thank the reviewer again. The fitting process in our model occurred only at the first stage where the synaptic parameters were fit to the Mooney and Prather as well as the Kosche results. There was no data shared and we merely looked at the figures in those papers and checked the amplitude of the elicited currents, the magnitudes of DC-evoked excitations etc … and we replicated that in our model. While this is suboptimal, it was better for us to start with it rather than simply using equations for synaptic currents from the literature for other types of neurons (that are not even HVC’s or in the songbird) and integrate them into our network model. The number of ODEs that govern the dynamics of every model neuron is listed on page 10 of the manuscript as well as in the Appendix. Moreover, we highlighted the details of this fitting process in the revised version.

Also disappointing is that the authors do not make any predictions to test, except rather weak ones such as that varying a maximum conductance sufficiently (which might be possible by using dynamic clamps) might cause burst propagation to stop or change its properties. Based on their results, the authors do not make suggestions for further experiments or calculations, but they should.

We agree that making experimental testable predictions is crucial for the advancement of the model. Our predictions include testing whether eradication of a class of neurons such as HVC_X_ neurons disrupts activity propagation which can be done through targeted neuron elimination. This also can be done through preventing rebound bursting in HVC_X_ by pharmacologically blocking the I_H_ channels. Others include down regulation of certain ion channels (pharmacologically done through ion blockers) and testing which current is fundamental for song production (and there a plenty of test based our results, like the SK current, the T-type Ca^2+^ current, the A-type K^+^ current, etc…). We incorporated these into the Discussion of the revised manuscript to better demonstrate the model's applicability and to guide future research directions.

Main issues:(1) Parameters are overly fine-tuned and often do not match known biology to generate chains. This fine-tuning does not reveal fundamental insights.(1a) Specific conductances (e.g. AMPA) are finely tweaked to generate bursts, in part due to a lack of a dendritic mechanism for burst generation. A dendritic mechanism likely reflects the true biology of HVC neurons.We acknowledge that the model does not include active dendritic processes and we do not regard this as a limitation. In fact, our present approach, although simplified, is intended to focus on somatic mechanisms to identify minimal conditions required for stable sequential propagation. We know HVC_RA_ neurons possess thin, spiny dendrites which can contribute to burst initiation and shaping. Future models that include such nonlinear dendritic mechanisms would likely reduce the need for fine tuning of specific conductances at the soma and consequently better match the known biology of HVC_RA_ neurons.

In text: “While our simplified, somatically driven architecture enables better exploration of mechanisms for sequence propagation, future extensions of the model will incorporate dendritic compartments to more accurately reflect the intrinsic bursting mechanisms observed in HVC_RA_ neurons.”

(1b) In this paper, microcircuits are simulated and then concatenated to make the HVC chain, resulting in no representations during silent gaps. This is out of touch with the known HVC function. There is no anatomical nor functional evidence for microcircuits of the kind discussed in this paper or in the earlier and rather similar paper by Eve Armstrong and Henry Abarbanel (J. Neurophy 2016). One can write a large number of papers in which one makes arbitrary unconstrained guesses of network structure in HVC and, unless they reveal some novel principle or surprising detail, they are all going to be weak.

Although the model is composed of sequentially activated microcircuits, the gaps between each microcircuit’s output do not represent complete silence in the network. During these periods, other neurons such as those in other microcircuits may still exhibit bursting activity. Thus, what may appear as a 'silent gap' from the perspective of a given output microcircuit is, in fact, part of the ongoing background dynamics of the larger HVC neuron network. We fully acknowledge the reviewer's point that there is no direct anatomical or physiological evidence supporting the presence of microcircuits with this structure in HVC. Our intention was not to propose the existence of such a physical model but to use it as a computational simplification to make precise sequential bursting activity feasible given the biologically realistic neuronal dynamics used. Hence, our use of 'microcircuits' refers to a modeling construct rather than a structural hypothesis. Even if the network topology is hypothetical, we still believe that the temporal structuring suggested allows us to generate specific predictions for future work about burst timing and neuronal connections.

(1c) HVC interneuron discharge in the author's model is overly precise; addressing the observation that these neurons can exhibit noisy discharge. Real HVC interneurons are noisy. This issue is critical: All reviewers strongly recommend that the authors should, at the minimum in a revision, focus on incorporating HVC-I noise in their model.

We agree that capturing the variability in interneuron bursting is critical for biological realism. In our model, HVC interneurons receive stochastic background current that introduces variability in their firing patterns as observed in vivo. This variability is seen in our simulations and produces more biologically realistic dynamics while maintaining sequence propagation. We clarify this implementation in the Methods section.

(1d) Address the finding that Kosche et al show that even with reduced inhibition, HVCra neuronal timing is preserved; it is the burst pattern that is affected.

The differences between the Kosche et al. (2015**)** findings and the predictions of our model arise from differences in the aspect of HVC function we are modeling. Our model is more sensitive to inhibition, which is a designed mechanism for achieving precise song patterning. This is a modeling simplification we adopted to capture specific characteristics of HVC function.

We acknowledged this point in the discussion: “While findings of Kosche et al. (2015) emphasize the robustness of the HVC timing circuit to inhibition, our model is more sensitive to inhibition, highlighting that HVC likely operates with several**,** redundant mechanisms that overall ensure temporal precision**.”**

(1e) The real HVC is robust to microlesions, cooling, and HVCra neuron turnover. The model in this paper relies on precise HVCra connectivity and is not robust.

Although our model is grounded in the biologically observed behavior of HVC neurons in vivo, we don’t claim that it fully captures the resilience seen in the HVC network. Instead, we see this as a simplified framework that helps us explore the basic principles of sequential activity. In the future, adding features like recurrent excitation, synaptic plasticity, or homeostatic mechanisms could make the model more robust.

(1f) There is unclear motivation for Ih-driven HVCx bursting, given past findings from the Mooney group.

Daou et al (2013) noticed that the observed in HVC_X_ and HVC_INT_ neurons in response to hyperpolarizing current pulses (Dutar et al. 1998; Kubota and Saito 1991; Kubota and Taniguchi 1998) was completely abolished after the application of the drug ZD 7288 in all of the neurons tested indicating that the sag in these HVC neurons is due to the hyperpolarization-activated inward current (I_h_). in addition, the sag and the rebound seen in these two neuron groups were larger as for larger hyperpolarization current pulses.

(1g) The initial conditions of the network and its activity under those conditions, as well as the possible reliance on external inputs, are not defined.

In our model, network activity is initiated through a brief, stochastic excitatory input to a small HVC_RA_ neuron of one microcircuit. This drive represents a simplified version of external input from upstream brain regions known to project to HVC, such as nuclei in the high vocal center's auditory pathways such as Nif and Uva. Modeling the activity of these upstream regions and their influence on HVC dynamics is an ongoing research work to be published in the future.

(1h) It has been known from the time of Hodgkin and Huxley how to include temperature dependences for neuronal dynamics so another suggestion is for the authors to add such dependences for the three classes of neurons and see if their simulation causes burst frequencies to speed up or slow down as T is varied.

We added this as limitation to the discussion section: “Our model was run at a fixed physiological temperature, but it's well known going all the way back to Hodgkin and Huxley that both ion channel activity and synaptic dynamics can change with temperature. In future work, adding temperature scaling (like Q10 factors) could help us explore how burst timing and sequence speed change with temperature changes, and how neural activity in HVC would/would not preserve its precision under different physiological conditions.”

(2) The scope of the paper and its objectives must be clearly defined. Defining the scope and providing caveats for what is not considered will help the reader contextualize this study with other work.(2a) The paper does not consider the role of external inputs to HVC, which are very likely important for the capacity of the HVC chain to tile the entire song, including silent gaps.

The role of afferent input to HVC particularly from nuclei such as Uva and Nif is critical in shaping the timing and initiation of HVC sequences throughout the song, including silent intervals. In fact, external inputs are likely involved in more than just triggering sequences, they may also influence the continuity of activity across motifs. However, in this study, we chose to focus on the intrinsic dynamics of HVC as a step toward understanding the internal mechanisms required for generating temporally precise sequences and for this reason, we used a simplified external input only to initiate activity in the chain.

(2b) The paper does not consider important dendritic mechanisms that almost certainly facilitate the all-or-none bursting behavior of HVC projection neurons. the authors need to mention and discuss that current-clamped neuronal response - in which an electrode is inserted into the soma and then a constant current-step is applied - bypasses dendritic structure and dendritic processing and so is an incomplete way to characterize a neuron's properties. In particular, claiming to fit current-clamp data accurately and then claiming that one now has a biophysically accurate network model, as the authors do, is greatly misleading.

While we addressed this is 1a, we do not suggest that our model is a fully accurate biophysical representation of HVC network. Instead, we see it as a simplified framework that helps reveal how much of HVC’s sequential activity can be explained by somatic properties and synaptic interactions alone. However, additional biological mechanisms, like dendritic processing, are likely to play an important role and should be explored in future work.

(2c) The introduction does not provide a clear motivation for the paper - what hypotheses are being tested? What is at stake in the model outcomes? It is not inherently informative to take a known biological representation and fine-tune a limited model to replicate that representation.

We explicitly added the hypotheses to the revised introduction.

(2d) There have been several published modeling efforts applied to the HVC chain (Seung, Fee, Long, Greenside, Jin, Margoliash, Abarbanel). These and others need to be introduced adequately, and it needs to be crystal clear what, if anything, the present study is adding to the canon.

While several influential models have explored how HVC might generate sequences ranging from synfire chains to recurrent dynamics or externally driven sequences (e.g., Seung, Fee, Long, Greenside, Jin, Abarbanel, and others), these models could not capture the detailed dynamics observed in vivo. Our aim was to bridge a gap in the modeling literature by exploring how far biophysically grounded intrinsic properties and experimentally supported synaptic connections that are local to the HVC can alone produce temporally precise sequences. We have proven that these mechanisms are sufficient to generate these sequences, although some missing components (such as dendritic mechanisms or external inputs) might be needed to fully capture the complexity and robustness of HVC function.

(2e) The authors mention learning prominently in the abstract, summary, and introduction but this paper has nothing to do with learning. Most or all mentions of learning should be deleted since they are misleading.

We appreciate the reviewer’s observation however our intent by referencing learning was not to suggest that our model directly simulates learning processes, but rather to place HVC function within the broader context of song learning and production, where temporal sequencing plays a fundamental role. Yet, repeated references to learning may be misleading given that our current model does not incorporate plasticity, synaptic modification, or developmental changes. Hence, we have carefully revised the manuscript to rephrase mentions of learning unless directly relevant to context.

(3) Using the model for hypothesis generation and prediction of experimental results.(3a) The utility of a model is to provide conceptual insight into how or why the real HVC functions as it does, or to predict outcomes in yet-to-be conducted experiments to help motivate future studies. This paper does not adequately achieve these goals.

We revised the Discussion of the manuscript to better emphasize potential contributions and point out many experiments that could validate or challenge the model’s predictions. These include dynamic clamp or ion channel blockers targeting A-type K^+^ in HVC_RA_ neurons to assess their impact on burst precision, optogenetic disruption of inhibitory interneurons to observe changes in burst timing and sequence propagation, pharmacological modulation of I_h_ or I_CaT_ in HVC_X_ and interneurons etc.

(3b) Additionally, it can be interesting to conduct an experiment on an existing model; for example, what happens to the HVCra chain in your model if you delete the HVCx neurons? What happens if you block NMDA receptors? Such an approach in a modeling paper can help motivate hypotheses and endow the paper with a sense of purpose.

We agree that running targeted experiments to test our computational model such as removing an HVC neuron population or blocking a synaptic receptor can be a powerful way to generate new ideas and guide future experiments. While we didn’t include these specific tests in the current study, the model is well suited for this kind of exploration. For instance, removing interneurons could help us better understand their role in shaping the timing of HVC_RA_ bursts. These are great directions for future experiments, and we now highlight this in the discussion as a way the model could be used to guide experiments.

(4) Changes to the paper's organization may improve clarity.(4a) Nearly all equations should be moved to an Appendix so that the main part of the paper can focus on the science: assumptions made, details of simulations, conclusions obtained, and their significance. The authors present many equations without discussion which weakens the paper.

Equations moved to appendix.

(4b) There are many grammatical errors, e.g., verbs do not match the subject in terms of being single or plural. The authors need to run their manuscript through a grammar checker.

Done.

(4c) Many of the figures are poorly designed and should be substantially modified. E.g. in Figure 1B, too many colors are used, making it hard to grasp what is being plotted and the colors are not needed. Figures 1C and 1D are entire figures taken from other papers, and there is no way a reader will be able to see or appreciate all the details when this figure is published on a single page. Figure 2 uses colors for dots that are almost identical, and the colors could be avoided by using different symbols. Figure 5 fills an entire page but most of the figure conveys no information, there is no need to show the same details for all 120 neurons, just show the top 1/3 of this figure; the same for Figure 7, a lot of unnecessary information is being included. Figure 10, the bottom time series of spikes should be replaced with a time series of rates, cannot extract useful information.

Adjusted as requested.

(4d) Table 1 is long and largely uninteresting, and should be moved to an appendix.

Table 1 moved to appendix.

(4e) Many sentences are not carefully written, which greatly weakens the paper. As one typical example, the first sentence in the Discussion section "In this study, we have designed a neural network model that describes [sic] zebra finch song production in the HVC." This is inaccurate, the model does not describe song production, it just explores some properties of one nucleus involved with song production. Just one or few sentences like this is ok but there are so many sentences of this kind that the reader loses faith in the authors.

Thank you for raising this point, we revised the manuscript to improve the precision of the writing. We replaced the first sentence of the discussion with this: "In this study, we developed a biophysically realistic neural network model to explore how intrinsic neuronal properties and local connectivity within the songbird nucleus HVC may support the generation of temporally precise activity sequences associated with zebra finch song."